# ATP plays a structural role in Hsp90 function

Michael Reidy ⬥ ✉ & Daniel C. Masison

Hsp90 is a highly conserved ATP-dependent molecular chaperone that forms a clamp around client proteins. The role of ATP in Hsp90 function is unclear since cell viability requires ATP binding, but not hydrolysis. Here, we present findings that support our hypothesis that after ATP binds, the γ phosphate repositions in a regulated manner to interact with a conserved arginine (R380) and stabilize the closed clamp. We propose that the essential role of ATP in Hsp90 function is structural: ATP is a linker that physically tethers the N and M domains and stabilizes closing. Severing this link by hydrolysis facilitates reopening. Our findings support the idea that R380 is an arginine finger, a motif found in diverse NTPase families, due to its interdomain interaction with ATP. This in turn suggests that for some arginine fingers the nucleotide itself is a structural element important for stabilization of inter-domain or -subunit interactions.

The 90 kilodalton heat shock protein (Hsp90) is an abundant and highly conserved molecular chaperone that is essential for eukaryotic life. Hsp90 regulates the activity of many different "client" proteins, assisted by a less-well conserved cohort of helper proteins called co-chaperones. Hsp90 clients vary widely in terms of protein family or class, and thus structure. Regulation of key cellular pathways means that Hsp90 plays a role in many important cellular processes, and therefore has become an attractive target for cancer and neurodegenerative disease therapies[1–4].

Hsp90 has three domains: the amino terminal "N" domain binds nucleotides, the middle "M" domain is the site of most client interactions and the carboxy terminal "C" domain mediates homodimerization. A highly flexible charged linker separates the N and M domains. Hsp90 is a member of the G̲yrB/H̲sp90/ histidine k̲inase/MutL̲ (GHKL) ATPase superfamily[5]. Hsp90 undergoes dramatic conformation rearrangements that are largely influenced by the nucleotide state of the N domain, the bound client, and certain co-chaperones. In the apo or ADP state Hsp90 is in the open conformation, a V-shape dimer joined at the C domains with the N domains of the two protomers far apart[6]. After ATP binding (but not strictly due to ATP binding per se, as discussed below), Hsp90 forms the so-called "closed clamp" around clients[7,8]. It is generally accepted that ATP hydrolysis occurs in the closed clamp conformation[9,10]. In the closed clamp, the N domains dock onto the M domains, and the N domains of the two protomers contact each other[11,12]. This docking results in the repositioning of a conserved arginine residue (R380 in *Saccharomyces cerevisiae* Hsp90,

called Hsp82) in the M domain to contact the γ-phosphate of ATP residing in the N domain[7,13]. The N-terminal β-strands of the N domains swap over each other, stabilizing the closed clamp[7]. Hydrolysis of ATP to ADP reopens the dimer and releases the bound client[14,15].

Although the relationship between Hsp70 - a universal in vivo interactor of Hsp90 - and ATP binding, hydrolysis and protein refolding was well defined about a decade after the discovery of the heat shock proteins[16–18], a comprehensive understanding of the relationship between Hsp90 and ATP has been persistently elusive[10]. Indeed, for several years, Hsp90's ability to even bind ATP was debated, due to a combination of Hsp90's very low affinity for ATP and its position within the nucleotide binding pocket that prevented Hsp90 from interacting with immobilized ATP reagents then available[19–23]. Hsp90's ability to bind ATP was eventually confirmed by X-ray crystallography and was observed to be in the same pocket that the Hsp90-specific inhibitor geldanamycin bound[24,25]. In 1998 Panaretou et al. finally observed Hsp90's ATP hydrolysis activity using the pyruvate kinase/lactate dehydrogenase (PK/LDH) coupled assay[26], which rapidly removes ADP (a potent inhibitor of Hsp90 ATPase[27]) from the reaction. In addition to confirming Hsp90's ATPase activity, Panaretou et al. concluded that both ATP binding and hydrolysis were required for essential Hsp90 function in *Saccharomyces cerevisiae*, by engineering mutations in Hsp82 that either blocked ATP binding (D79N) or hydrolysis (E33A)[26]. Yeast cells expressing either mutant as the sole source of Hsp90 failed to grow. Later the same year, the Hartl group reported similar findings as Panaretou et al. using a very similar system[28].

Laboratory of Biochemistry and Genetics, National Institute of Diabetes and Digestive and Kidney Diseases, National Institutes of Health, Bethesda, MD, USA. ✉e-mail: michael.reidy@nih.gov

The idea that Hsp90 ATP hydrolysis is essential for viability has become dogma and is the lens through which all observations are seemingly interpreted. This view is reasonable given that we know of no other essential protein that must bind ATP but not hydrolyze it to function. However, more recent reports have demonstrated that Hsp90 indeed need not be able to hydrolyze ATP to support viability, as cells expressing the E33A variant of Hsp90 were found to be viable by two different groups[29,30]. The specific reason for the conflict between earlier and recent reports of E33A viability is unclear. While the E33A mutation was found to severely impair yeast cell growth and disrupt certain Hsp90-regulated pathways[29,30], secondary mutations in the E33A mutant protein were identified that rescued E33A-mediated growth defects but did not restore ATP hydrolysis[30]. Furthermore, the dispensability of Hsp90's ATPase and the effect of the suppressing secondary mutations were conserved, as hydrolysis-defective Hsp90s from several species, including both human Hsp90 isoforms, supported viability in *S. cerevisiae* and the equally distantly related (compared to both human and *S. cerevisiae*) *Schizosaccharomyces pombe*[30]. Therefore, the energy of ATP hydrolysis is not required for the maturation of Hsp90's clients, and ATP must be doing something that is needed for proper Hsp90 function besides providing energy for protein refolding. This idea is supported by recent reports of ATPase-independent activities of various Hsp90 orthologs[31–36]. Taken together, these observations indicate that the structural movements required for dimer opening associated with ATP hydrolysis[15] must, at some threshold level, be replicated by ATP-to-ADP exchange in vivo.

The discrepancies between the more recent and older studies show that questions regarding Hsp90's relationship with ATP require renewed attention.

Here, we report on the effects of mutating residues in or near the nucleotide binding pocket on the genetic, biochemical and biophysical properties of Hsp90, to better understand the role of ATP in Hsp90 function. Our results provide further evidence that ATPase is not correlated to viability, but rather suggest that the ability to form the closed clamp determines whether a mutant can support cell growth. In addition, our findings reinforce the recently proposed idea that after binding, ATP must adopt a specific orientation to contact R380 and stabilize the closed clamp[30]. Taking the results presented here together with the findings of many groups spanning decades of Hsp90 research, we propose that a role of ATP in Hsp90 function is structural: it is a severable linker that facilitates both the formation and dissociation of the closed clamp via processes that can be regulated by co-chaperones, clients and other factors.

## Results

### Positive charge at position 380 is necessary and sufficient for closed clamp stabilization and in vivo Hsp90 functions

Purified Hsp90 exists in an equilibrium of open and closed conformations. In the apo or ADP state the equilibrium is shifted mostly toward the open conformation[11]. We use a photon-induced electron transfer (PET) assay developed by Schulze et al. to observe docking of the N and M domains, which serves as a proxy for formation of the closed clamp (See "Methods" and Fig. 1a)[11]. Addition of non-hydrolysable ATP analogs such as AMPPNP or ATP-γ-S to wild type Hsp82 rapidly shifts the equilibrium from mostly open to mostly closed, observed as fluorescence quenching of appropriately labeled proteins (Fig. 1b, left)[11,30,37]. We and others have shown that addition of ATP to purified wild type Hsp82 does not induce formation of the closed clamp like the non-hydrolysable analogs (Fig. 1b, left)[11,27,29,30,38–40]. Yet, in all likelihood ATP induces closing in vivo since mutations that block either closing or ATP binding are lethal[26,28–30].

We recently reported that the E33A mutation "flipped" the effects the different nucleotides had on Hsp82 conformation, results that we confirm here (Fig. 1b, right): the non-hydrolysable analogs had no effect on conformation of Hsp82$^{E33A}$ while ATP stabilized the closed

clamp[30]. To explain these observations, we proposed that for wild type Hsp90 to form the closed clamp, ATP must adopt an orientation that positions the γ-phosphate near R380 in the M domain so that they can interact. Binding of R380 to the γ-phosphate stabilizes N-M domain docking and thus the closed clamp[13] (Fig. 1c). This idea is supported by the observation that the γ-phosphate of ATP is not visible, and is thus mobile, in the crystal structure of the Hsp90 N domain, which resembles the N domain in the open state of full length Hsp90 where the N and M domains are separated[24]. We further proposed that the altered geometry or mobility of the triphosphate tails of the non-hydrolysable analogs pre-positioned the γ-imidophosphate or γ-thiophosphate (of AMPPNP or ATP-γ-S, respectively) or made the transition to the orientation needed to contact R380 easier than the γ-phosphate of ATP (Fig. 1d). We postulated that E33A altered the shape of the nucleotide binding pocket in a way that flipped the responses to the different nucleotides[30].

R380 is conserved among GHKL ATPase superfamily members[5], and early reports suggested a catalytic role for R380[41]. This idea was supported by the visualization of the R380:γ-phosphate interaction in the crystal structure of the Hsp82 closed clamp[7]. In 2012 it was reported that AMPPNP did not stabilize the closed conformation of the Hsp82$^{R380A}$ mutant using small angle X-ray scattering[13]. The authors of that study noted that Hsp82$^{R380A}$ could nevertheless hydrolyze ATP and concluded that R380 was not catalytic but was an "ATP sensor" that stabilized N-M docking to facilitate ATP hydrolysis[13]. Later studies reported similar results[15,29]. We reasoned that if the electrostatic interaction between the γ-phosphate of ATP and R380 was crucial for formation of the closed-clamp, then a lysine substitution might behave similarly to wildtype in the N-M docking assay. We used Hsp82$^{R380A}$ as a control and found that it was unable to form the closed clamp with any nucleotide as determined by the N-M docking PET assay, in agreement both with the earlier studies[13,29], and with the idea that γ-phosphate-R380 interaction is needed to stabilize the closed clamp (Fig. 1e, left). In line with our prediction, Hsp82$^{R380K}$ responded to ATP-γ-S like wild type in the N-M docking assay (Fig. 1e, right). However, the R380K substitution altered the response to both AMPPNP (less docking than wild type) and ATP (more docking than wild type), while having no effect on ATPase rate (Fig. 1f). These results support our hypothesis that ATP adopts a specific orientation to induce N-M docking: the different chemical structure of lysine compared to arginine alters the location of the positive charge which in turn changes the γ-phosphate's ability to interact with it and stabilize the closed clamp.

We refer here to mutant Hsp82 proteins that can support viability as the sole source of Hsp90 as functional. We reasoned that since positive charge at position 380 is sufficient for closed clamp formation in the PET assay, then Hsp82$^{R380K}$ would function in vivo. To test this idea, we expressed the R380 mutants as the only source of Hsp90 in *S. cerevisiae* cells by employing a widely used counterselection technique (see "Methods"). As previously reported, Hsp82$^{R380A}$ was unable to support viability, since cells lacking chromosomal Hsp90 genes that were transformed with a *TRP1*-marked plasmid encoding Hsp82$^{R380A}$ could not lose the parental *URA3*-marked wild type Hsp90 plasmid during growth on uracil-containing medium and were subsequently killed by 5′-fluoro-orotic acid (FOA), which is toxic to cells expressing *URA3* (Fig. 1g; see Methods for details)[29]. Cells expressing Hsp82$^{R380K}$ were recovered from FOA (Fig. 1g) and had a similar growth rate as cells expressing wild-type Hsp82 (Fig. 1h), indicating that Hsp82$^{R380K}$ is fully functional. Furthermore, Hsp82$^{R380K}$ in vivo function was not dependent on the Hsp90 co-chaperones Sti1, Cpr6/7, Aha1, Ppt1 or Sba1 (Supplementary Fig. 1). These results support the broadly held notion that the ability to form the closed clamp in vitro (via the γ-phosphate binding to positively charged residues at position 380) is strongly linked to the ability to function in vivo.

Hsp90 mutants that support viability may be defective in pathways that are non-essential under optimal growth conditions

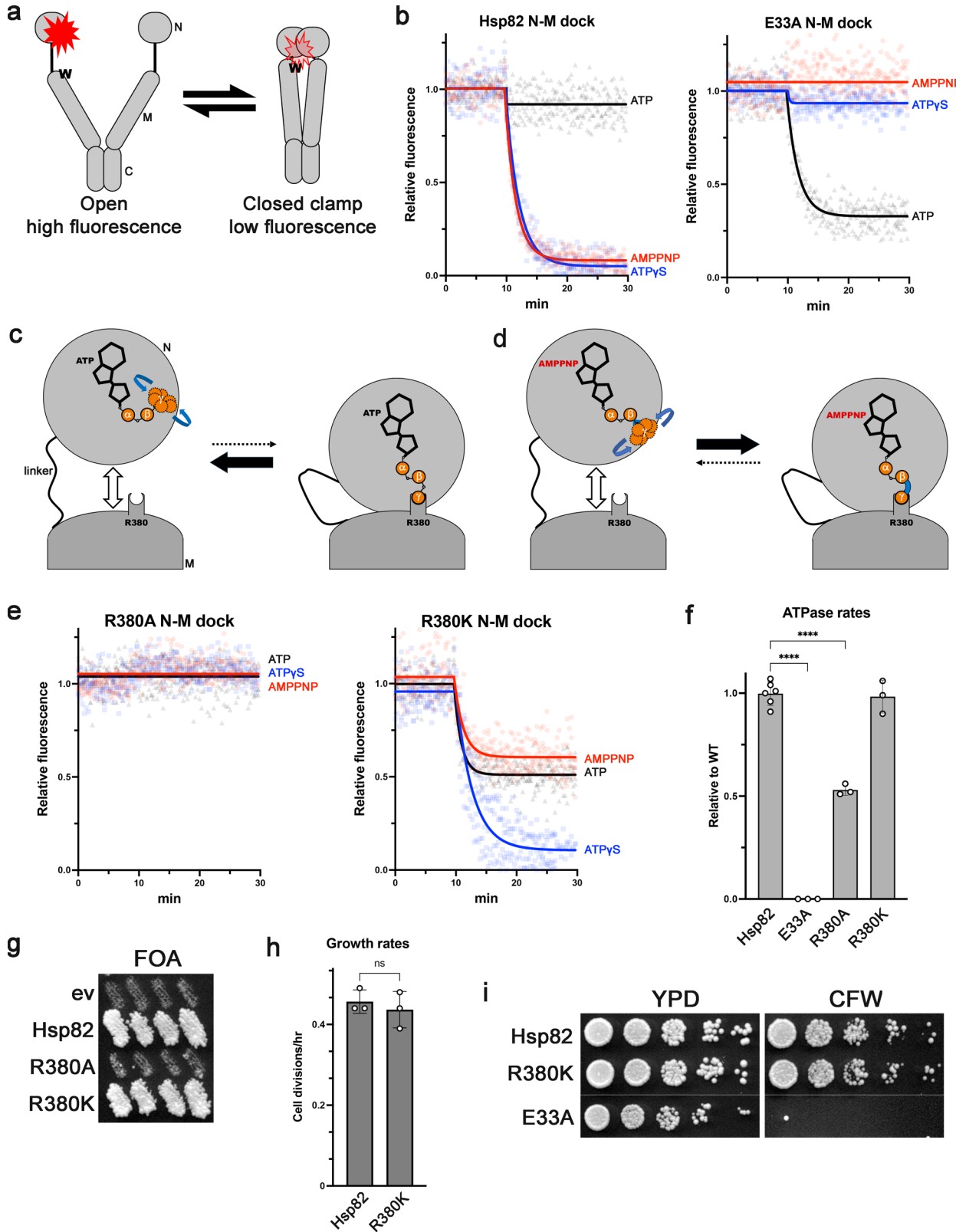

and thus would be considered partially functional. *S. cerevisiae* cells react to cell wall damage by activating the cell wall integrity (CWI) pathway, a MAP kinase cascade that transmits damage signals from the cell wall to transcription factors that activate expression of cell wall proteins and repair enzymes[42]. Several kinases in the CWI pathway are Hsp90 clients[43–46]. Sensitivity to cell wall damaging agents such as calcofluor white (CFW) thus reflects reduced ability

of Hsp90 mutants to chaperone kinase clients. The R380K mutation did not impair Hsp90's ability to regulate MAP kinase clients, since cells expressing Hsp82[R380K] grew like wild type on medium containing CFW (Fig. 1i). Taken together, these data strongly suggest that positive charge at position 380 in the M domain is necessary and sufficient for formation of the closed clamp and in vivo Hsp90 functions.

**Fig. 1 | Positive charge at position 380 in the M domain of Hsp90 is critical for closed clamp formation. a** Cartoon showing the PET experimental setup. Molecules in the open conformation (left) have high fluorescence. Molecules in the closed clamp conformation (right) have low fluorescence due to quenching of the fluorophore in the N domain by an engineered tryptophan (W) in the M domain. **b** The response of wild type Hsp82 (left) and Hsp82[E33A] to AMPPNP (red), ATP-γ-S (blue) and ATP (black) in the PET assay. Nucleotides were added to the proteins at T = 10 min. **c** Cartoon depicting the motion (blue arrows) of the ATP γ-phosphate in the open state (left) and bound to R380 in the closed clamp (right). **d** Like in c. except with AMPPNP. The γ imidophosphate of AMPPNP is structurally predisposed to contact R380 (compare to position of γ-phosphate in c.) and thus forms the closed clamp more readily than ATP (compare black arrows to c.). **e** The response of Hsp82[R380A] (left) and Hsp82[R380K] (right) to AMPPNP (red), ATP-γ-S (blue) and ATP (black) in the PET assay. **f** ATPase rates of Hsp82[E33A], Hsp82[R380A], and Hsp82[R380K]

relative to wild type Hsp82. Bars are the average rates of at least three reactions (each replicate is shown as a circle) and error bars are the standard deviation. Asterisks indicate significant difference from wild type ($P < 0.0001$ by one-way ANOVA). **g** In vivo Hsp90 functional assay (See "Methods"). Cells expressing Hsp82[R380A] were inviable while those expressing Hsp82[R380K] grew like wild type on plates containing FOA. **h** Cells expressing Hsp82[R380K] had a similar growth rate as cells expressing wild type Hsp82 (not significant by one-way ANOVA). Bars are the average growth rates of three biological replicates (each replicate is shown as a circle) and error bars are the standard deviation. **i** Cells expressing Hsp82[R380K] were able to chaperone kinase clients since they grew like wild type on calcfluor white (CFW, see text). Cells expressing Hsp82[E33A] are shown as a CFW sensitive control. Lines in the images indicate where portions were cropped from different plates of identical composition.

### Table 1 | Summary of in vivo and in vitro mutational analyses

| Mutant | In vivo | | | In vitro | | | | Ref |
|---|---|---|---|---|---|---|---|---|
| | Func[a] | Phenotype | DN[b] | AMPPNP[c] | ATP-γ-S[c] | ATP[c] | ATPase[d] | |
| WT | Yes | No | No | 100 | 100 | - | 1 | |
| E33A | Yes | Slow, CFW[S] | No | - | - | 80 | none[e] | 29,30 |
| R32A | No | n/a | No | - | - | - | 0.54 ± 0.02 | 47 |
| D40A | No | n/a | No | - | - | - | 0.15 ± 0.07 | 24 |
| K98A | Yes | No | No | 100 | 100 | 50 | 0.93 ± 0.03 | 27,30 |
| G118A | No | n/a | Yes | > open[f] | > open | 80 | none | 24 |
| F120A | Yes | Slow, CFW[S] | No | - | 100 | 50 | 1.29 ± 0.02 | 7,8,53 |
| G121A | No | n/a | No | - | - | - | 0.03±0.02 | 24 |
| G123A | No | n/a | Yes | >open | 100 | 10 | 0.61 ± 0.02 | 24 |
| F124A | Yes | Slow, CFW[S] | No | 80 | 100 | 100 | 3.33 ± 0.07 | 24,76 |
| A152V | Yes | No | No | 100 | 100 | 100 | 1.80 ± 0.16 | 37 |
| R380A | No | n/a | No | - | - | - | 0.53 ± 0.03 | 13,15,29 |
| R380K | Yes | No | No | 50 | 100 | 50 | 0.98 ± 0.08 | |

[a] *func* function in vivo (support viability).

[b] *DN* dominant negative phenotype.

[c] *AMPPNP, ATP-γ-S, ATP* the approximate level of closed clamp stabilization induced by the nucleotide observed in PET assays, compared to wild type.

[d] *ATPase* rate relative to wild type.

[e] *none* ATP hydrolysis activity not detected.

[f] *>open* more open compared to the apo state of this mutant.

## Combined effects of nucleotide binding pocket mutation and nucleotide on Hsp90 closing dynamics

If E33A and R380K altered the contours of the nucleotide binding pocket in a way that changed the abilities of the different nucleotides to stabilize the closed clamp, then similar effects might be observed when other residues in the nucleotide binding pocket or near R380 are mutated. To test this idea, we made alanine substitutions at residues R32, D40, K98, G118, F120, G121, G123 or F124 which are near the bound nucleotide[24] and a valine substitution at A152 that was shown to enhance both closing and reopening of Hsp90[37]. The residues we chose to mutate are conserved and described in further detail below (Table 1).

First, we asked if the mutant Hsp82s could function in vivo by testing if they support viability of *S. cerevisiae* as the sole source of Hsp90. We found that the R32A, D40A, G118A, G121A and G123A substitutions of Hsp82 were lethal (Fig. 2a). Failure of these nonfunctional mutants to support viability was not due to lack of expression (Supplementary Fig. 2a). During the FOA counterselection procedure we observed that cells expressing both wild type Hsp82 and either Hsp82[G118A] or Hsp82[G123A] grew significantly slower than cells expressing both wild type Hsp82 and any of the other nonfunctional mutant Hsp82 proteins or an empty vector control (Fig. 2b). Thus, Hsp82[G118A] and Hsp82[G123A] have a dominant negative growth phenotype which suggests that they interfere with the function of wild type Hsp82.

We next tested the ability of the functional mutants to chaperone MAP kinase clients by monitoring sensitivity to CFW. Cells expressing Hsp82[F120A] or Hsp82[F124A] as the sole source of Hsp90 failed to grow on CFW (Fig. 2c). The reduced function of these mutants was also reflected in their slower growth rates (Supplementary Fig. 2b).

All of the mutants that supported growth (K98A, F120A, F124A, and A152V) also functioned in strains containing deletions of the Hsp90 co-chaperones Sti1, Aha1, Cpr6/7, Ppt1 or Sba1 (Supplementary Fig. 2c). Thus, the functional mutant proteins were not dependent upon the activities of these nonessential co-chaperones.

To determine whether the mutations affected the pattern of conformational response to the three adenine triphosphate nucleotides like E33A and R380K, we purified the mutant proteins and measured how they responded to ATP, AMPPNP or ATP-γ-S in the PET assay.

Residue R32 forms an ion pair with E33 and was recently shown to be important for proper conformational dynamics[47]. We observed that AMPPNP, ATP or ATP-γ-S failed to change the equilibrium of open and closed states of the nonfunctional Hsp82[R32A] protein (Fig. 2d). D40 and K98 are located near the ribose and α-phosphate moieties of the nucleotide, on opposite sides of the binding pocket relative to each other. None of the nucleotides induced nonfunctional Hsp82[D40A] to close (Fig. 2e). Hsp82[K98A], which functioned normally in vivo, responded to AMPPNP and ATP-γ-S like wild type and ATP partially stabilized

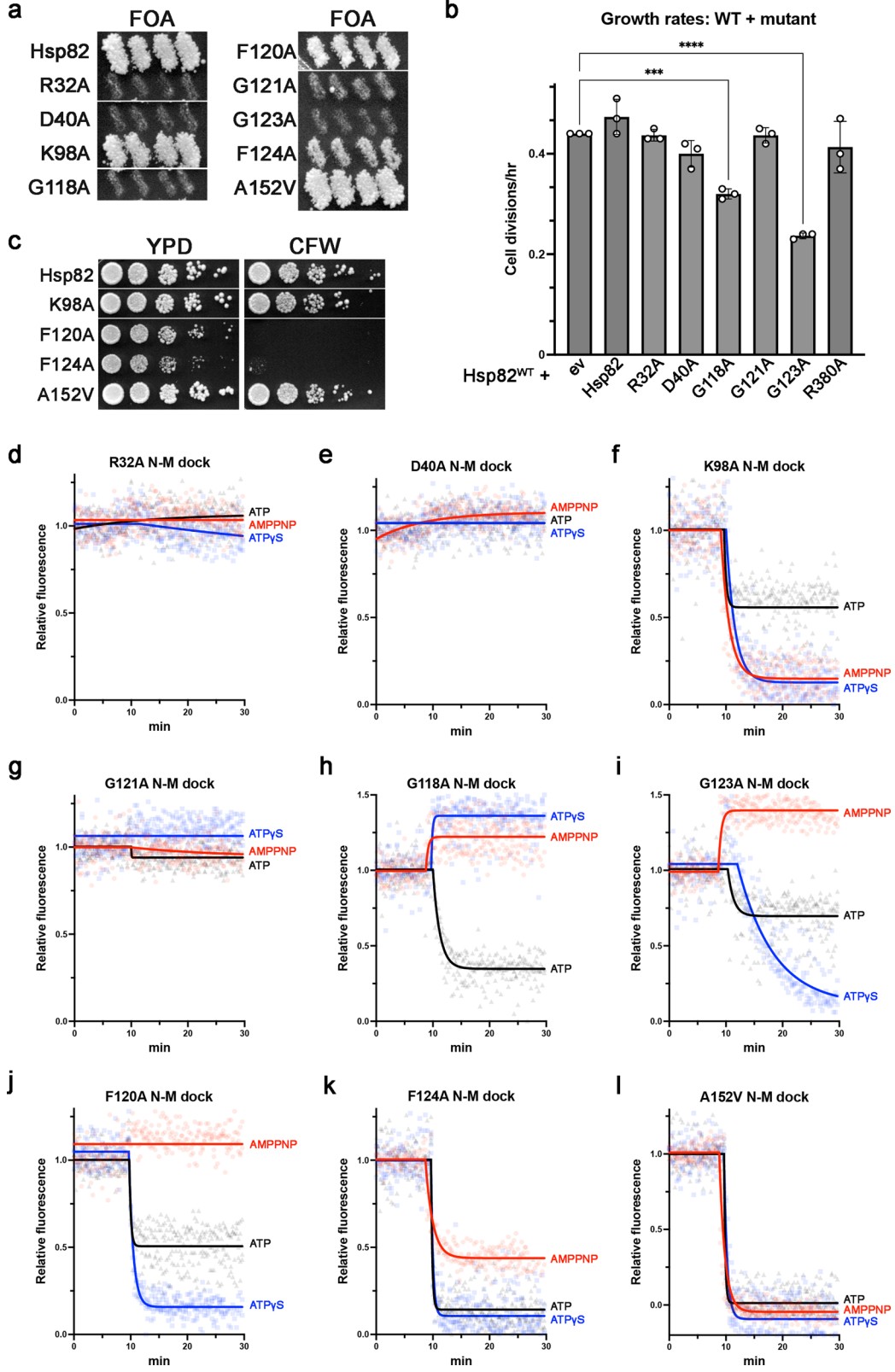

**Fig. 2 | Altering the nucleotide binding pocket landscape changes the ability of ATP and ATP analogs to induce the closed clamp. a** Cells expressing Hsp82$^{R32A}$, Hsp82$^{D40A}$, Hsp82$^{G118A}$, Hsp82$^{G121A}$, or Hsp82$^{G123A}$ were not viable, as evidenced by lack of growth on FOA plates. **b** Growth rates of cells expressing both wild type Hsp82 and the indicated Hsp82 mutant or empty vector ("ev") control. Bars are the average growth rates of three biological replicates (each replicate is shown as a circle) and error bars are the standard deviation. Asterisks indicate significant difference from wild type (*** $p < 0.001$, **** $p < 0.0001$ by one-way ANOVA). **c** Cells expressing Hsp82$^{F120A}$ or Hsp82$^{F124A}$ were defective in kinase client chaperoning as evidenced by CFW sensitivity. For (**a**, **c**) lines in the images indicate where portions were cropped from different areas of the same plate. **d**–**l** The response of the indicated Hsp82 mutant to AMPPNP (red), ATP-γ-S (blue) and ATP (black) in the PET assay. Nucleotides were added to the proteins at T = 10 min.

the closed clamp (Fig. 2f). Thus, mutations in the ribose-α-phosphate area of the nucleotide binding pocket have different effects on γ-phosphate-R380 interaction depending on their location.

We next investigated the role of three conserved glycine residues in γ-phosphate-R380 interaction. G118, G121 and G123 coordinate water molecules that interact with the triphosphate tail of the nucleotide[24]. In the PET assay, none of the nucleotides stabilized closed clamp formation of nonfunctional Hsp82[G121A] (Fig. 2g). In contrast to the other nonfunctional mutant proteins, Hsp82[G118A] and Hsp82[G123A], which had dominant negative effects (Fig. 2b), were able to form the closed clamp with at least one of the nucleotides. ATP stabilized the closed clamp of Hsp82[G118A] and had a modest effect on Hsp82[G123A] (Fig. 2h, i, black lines). AMPPNP shifted both Hsp82[G118A] and Hsp82[G123A] to a more open state (Fig. 2h, i, red lines). ATP-γ-S also shifted Hsp82[G118A] to a more open state but stabilized the closed clamp of Hsp82[G123A] (Fig. 2h, i, blue lines). Thus, these glycines and the water molecules they coordinate seem to be crucial for proper γ-phosphate-R380 interaction since mutating any of them resulted in very different effects compared to wild type on closed clamp stabilization by the three nucleotides.

In the closed clamp, F120 extends into the M domain and resides very close to R380 in contact with the γ-phosphate. We observed that AMPPNP had little effect on the partially functional Hsp82[F120A] while ATP-γ-S stabilized the closed state and ATP moderately induced closing (Fig. 2j). Thus, F120 may be important for proper closing dynamics by sterically regulating the position of R380. Residues F124 and A152 are located further from the triphosphate tail than the other residues in our panel. The phenyl sidechain of F124 is close to the adenosine ring of the nucleotide while A152 is on the surface of the N domain. The closed clamp was stabilized in both partially functional Hsp82[F124A] (Fig. 2k) and fully functional Hsp82[A152V] (Fig. 2l) by all three nucleotides, demonstrating that even changes outside the area of the triphosphates can impact γ-phosphate-R380 interaction.

Measurement of the mutant proteins' ATPase rates (Supplementary Fig. 2d) revealed that ATP hydrolysis did not correlate to a mutant's ability to support viability. Furthermore, the mutant's ATPase rates also could not explain differences in growth rates of viable strains, as mutations that elevated ATPase (F120A, F124A and A152V) either had no effect on growth (A152V) or caused cells to grow slowly and be CFW sensitive (F120A and F124A). Finally, the effects of the mutations on ATPase rates were not reflected in the PET assays, since Hsp82[R32A], Hsp82[D40A] and Hsp82[R380A] did not close in response to any nucleotide but still hydrolyzed ATP.

The results of the mutation analysis experiments are summarized in Table 1. While our experimental designs intentionally minimized possible effects the mutations had on nucleotide affinity, we cannot rule out that some of our observations are influenced by alterations in affinity. Regardless, all mutant proteins that functioned in vivo formed the closed clamp with at least one nucleotide in vitro. Notably, although Hsp82[E33A], Hsp82[F120A] and Hsp82[F124A] each responded differently to the three nucleotides in the PET assays, they had similar growth defects, demonstrating a lack of correlation between the effect the mutations had on phenotype and the ability of the three nucleotides to stabilize closed clamp formation. All of the mutants that did not form the close clamp with any nucleotide were nonfunctional in vivo. Thus, the ability to close with any nucleotide in the PET assay correlates with in vivo function. Yet, the ability to form the closed clamp is not necessarily sufficient to function in vivo, as Hsp82[G118A] and Hsp82[G123A] could close in the PET assay but were nonfunctional in vivo. However, unlike the other nonfunctional mutants Hsp82[G118A] and Hsp82[G123A] had dominant negative growth defects.

### E372K rescues growth defects of R32A and E33A and restores closed clamp stabilization by ATP-γ-S in vitro

Recently we showed that a mutation at a conserved glutamate (E372K in yeast Hsp82) suppressed the growth and MAP kinase chaperoning

defects mediated by the hydrolysis defective E33A mutation, but did not restore ATP hydrolysis[30]. E372 is in the same loop as R380. We were interested to know if any of the reduced- or nonfunctional mutants in our panel could be rescued by E372K like E33A. First, we observed that E372K had no effect on the slow growth or CFW sensitivity caused by F120A or F124A (Supplementary Fig. 3). Of the nonfunctional mutants, only R32A was rescued by E372K (Fig. 3a), but cells expressing Hsp82[R32A,E372K] were very slow and CFW sensitive (Fig. 3b). This result was puzzling as Hsp82[R32A] exhibited very little to no closing in the PET assay (Fig. 2a and Table 1) and viability was correlated to the ability to form the closed clamp. We therefore considered that E372K overcame the inhibition to close caused by R32A. Indeed, we observed that the closed clamp of Hsp82[R32A,E372K] was more stabilized by ATP-γ-S than Hsp82[R32A] (Fig. 3c). Interestingly, the effect was specific to ATP-γ-S, since like Hsp82[R32A], Hsp82[R32A,E372K] was unresponsive to AMPPNP or ATP (Fig. 3c). These results align with and strengthen the idea that in vivo function is correlated with the ability to form the closed clamp.

We next tested the effect of combining E372K and E33A on closing dynamics and observed that ATP-γ-S increased the stability of the closed clamp of Hsp82[E33A,E372K] much more than Hsp82[E33A] (Fig. 3d). Thus, the restorative effect of E372K on closing dynamics was very similar when combined with either the R32A or E33A mutations. Furthermore, compared to R32A the magnitude of E372K's effect was greater when combined with E33A both in vivo and in vitro. Taken together, these results suggest that E372K overcomes alterations to the nucleotide binding pocket that block or modify the dynamics of closed clamp formation, perhaps by influencing the position of R380 as they are on the same loop. However, the suppressive effect of E372K is specific to mutations in the R32-E33 area of the nucleotide binding pocket, as it was unable to rescue the lethality or growth defects of the other mutants in our panel. These results lend further support to our idea that the shape of the nucleotide binding pocket influences the position of the γ-phosphate and thus stabilization of the closed clamp.

### Different co-chaperones influence Hsp90's conformational dynamics by either promoting or inhibiting ATP-R380 interaction

A recent report concluded that reducing ATP's degrees of freedom in the binding pocket of Hsp90 by Aha1 interaction or molecular crowding facilitated formation of the closed clamp[48]. This observation is in line with our theory that ATP adopts a specific conformation that stabilizes the closed clamp. We hypothesized that in vivo, co-chaperones may regulate Hsp90's conformational dynamics by influencing the position of ATP's γ-phosphate. We tested this idea by including Hsp90 co-chaperones Sti1 or Sba1 in the PET assay. Based on what is known of these co-chaperones, we expected that Sti1 would have little effect on ATP's ability to promote closing of Hsp90, since Sti1 stabilizes the open form of Hsp90[49–51], and that Sba1 would increase the ability of ATP to stabilize the closed clamp in the N-M docking PET assay[7,51].

As expected, the presence of Sti1 shifted the equilibrium of Hsp82 more towards the open state in both apo and ATP conditions (Fig. 4a). Addition of AMPPNP to Hsp82:Sti1 complexes induced closing but more slowly and less completely than when Sti1 was not present (Fig. 4a). A recent study reported similar findings using ATP-γ-S[50]. These results lend support to our idea that the non-hydrolysable analogs are structurally predisposed to induce closed clamp formation compared to ATP, as AMPPNP was able to partially overcome the inhibitory effect of Sti1 on closed clamp formation.

Sba1 shifted the equilibrium of Hsp82 towards the closed state upon addition of ATP to ~50% of the effect of AMPPNP on Hsp82 alone (Fig. 4b). We then tested the effect of the molecular crowding agent trimethylamine N-oxide (TMAO), which can mimic the crowded environment of the cell cytosol[52]. We observed that TMAO alone enhanced closing in response to ATP to a similar degree as Sba1

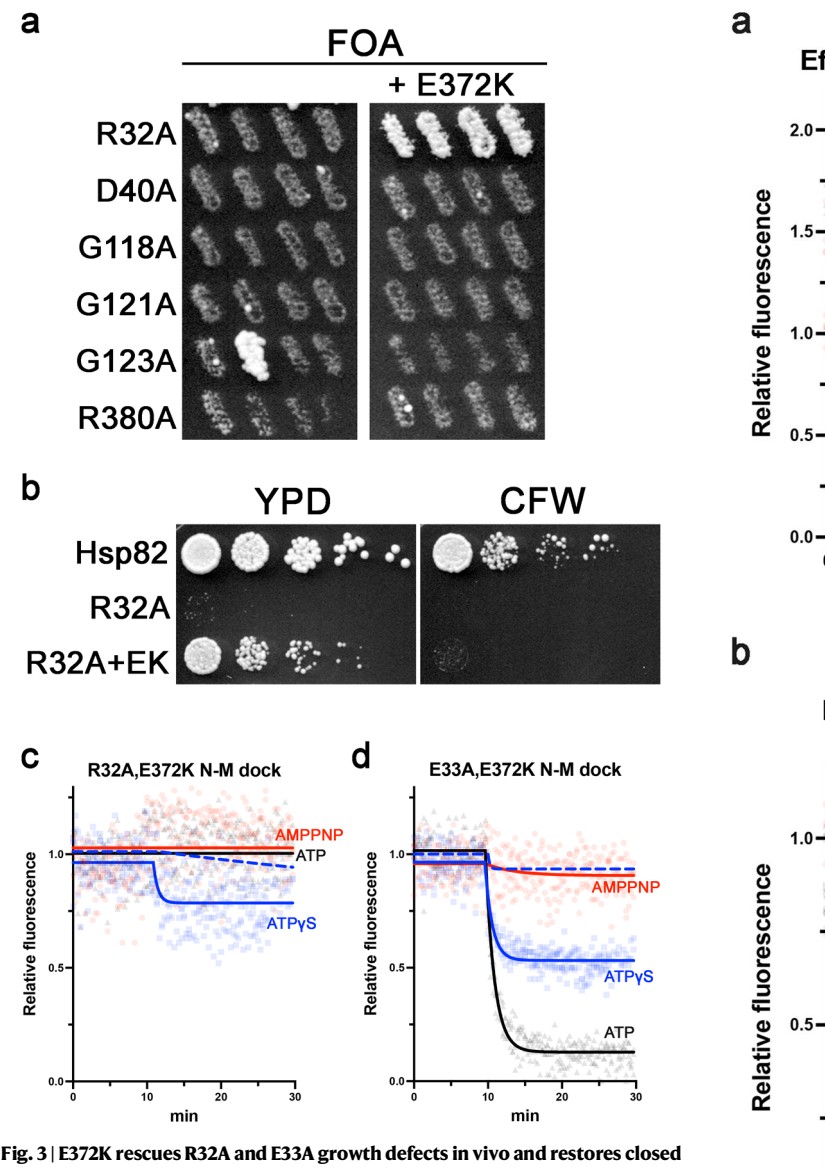

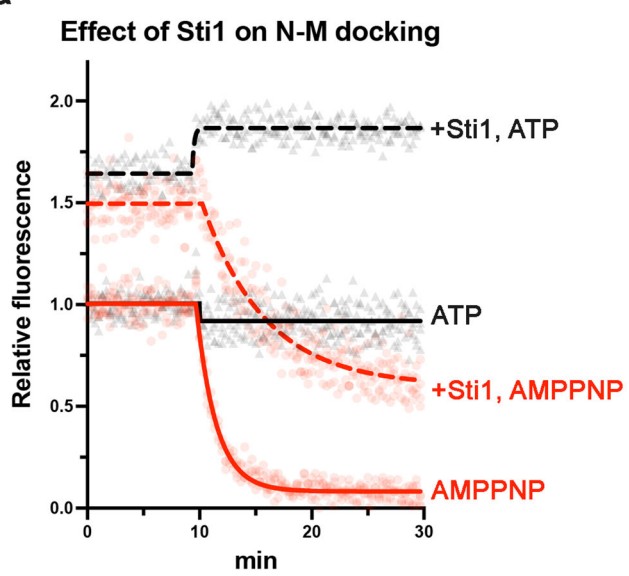

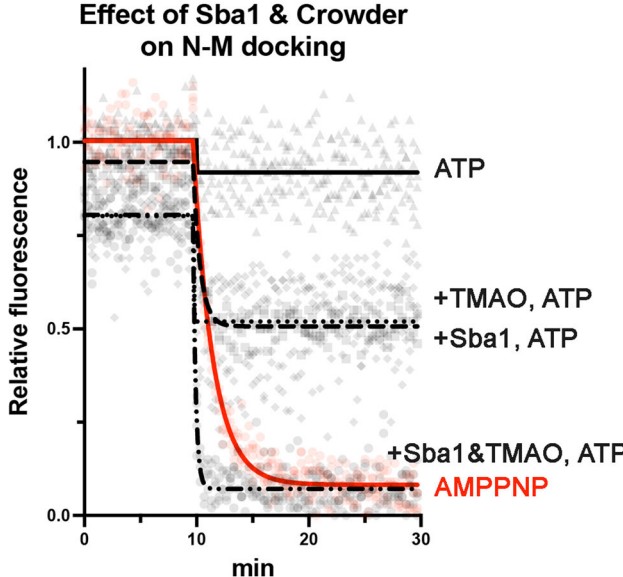

**Fig. 3 | E372K rescues R32A and E33A growth defects in vivo and restores closed clamp formation by ATP-γ-S in vitro. a** Hsp90 functional assay with the indicated inviable nucleotide binding pocket mutants (left column) and the same but combined with E372K. The E372K mutation rescued R32A but not the others. Growth of one replicate of G123A is an escape event and not true complementation, see "Methods". Both images were cropped from the same plate. **b** Cells expressing Hsp82[R32A,E372K] were viable but slow growing (left) and CFW sensitive (right). **c** The response of Hsp82[R32A,E372K] mutant to AMPPNP (red), ATP-γ-S (blue), and ATP (black) in the PET assay. Compare blue solid and dashed (Hsp82[R32A] plus ATP-γ-S, reproduced from Fig. 2a) lines. **d** The response of Hsp82[E33A,E372K] mutant to AMPPNP (red), ATP-γ-S (blue), and ATP (black) in the PET assay. Compare blue solid to dashed (Hsp82[E33A] plus ATP-γ-S, reproduced from Fig. 1b) lines. For **c**, **d** nucleotides were added to the proteins at T = 10 min.

**Fig. 4 | Co-chaperones influence Hsp90's conformational dynamics differently in response to ATP or AMPPNP. a** Responses of Hsp82 alone (solid lines) or with an equimolar (to Hsp90 dimers) amount of Sti1 (dashed lines) to AMPPNP (red) or ATP (black) in the N-M docking PET assay. **b** The responses of Hsp82 to ATP in the presence of Sba1 (dashed black line), TMAO (dotted black line) or both Sba1 and TMAO (dotted, dashed black line) in the PET assay. The response of Hsp82 alone to AMPPNP (red) is included as a reference. For (**a**, **b**) co-chaperones and/or crowding agent was preincubated with Hsp82 before T = 0.

(Fig. 4b). Combining Sba1 and the crowder resulted in a rate and level of closing of wild type Hsp82 with ATP that was comparable to AMPPNP without Sba1 and crowder (Fig. 4b). Thus, restriction of Hsp82's conformational fluctuations and/or ATP's movements within the binding pocket of Hsp82 allowed ATP to stabilize the closed clamp, in line with earlier reports and in support of our theory connecting the orientation of ATP with closed clamp formation[48]. This conclusion in turn suggests that co-chaperones and the cellular environment regulate formation of the closed clamp upon binding of ATP to wild type Hsp90 in vivo.

## Cdc37 regulates multiple aspects of Hsp90's function in vivo

Of the mutants we chose to investigate, F120 was particularly interesting because in the closed clamp the phenyl sidechain lies adjacent to R380, rather than interacting with ATP (Fig. 5a). As noted above, compared to wild type, ATP-γ-S and ATP (partially) stabilized the closed clamp of Hsp82[F120A], while AMPPNP had little effect on closing (Fig. 2j, Table 1). Due to its proximity to R380, we considered that loss of the bulky aromatic side chain in Hsp82[F120A] reduced restriction on R380's movement, resulting in the abnormal (compared to wild type) patterns observed in the PET assays. To test this idea and further

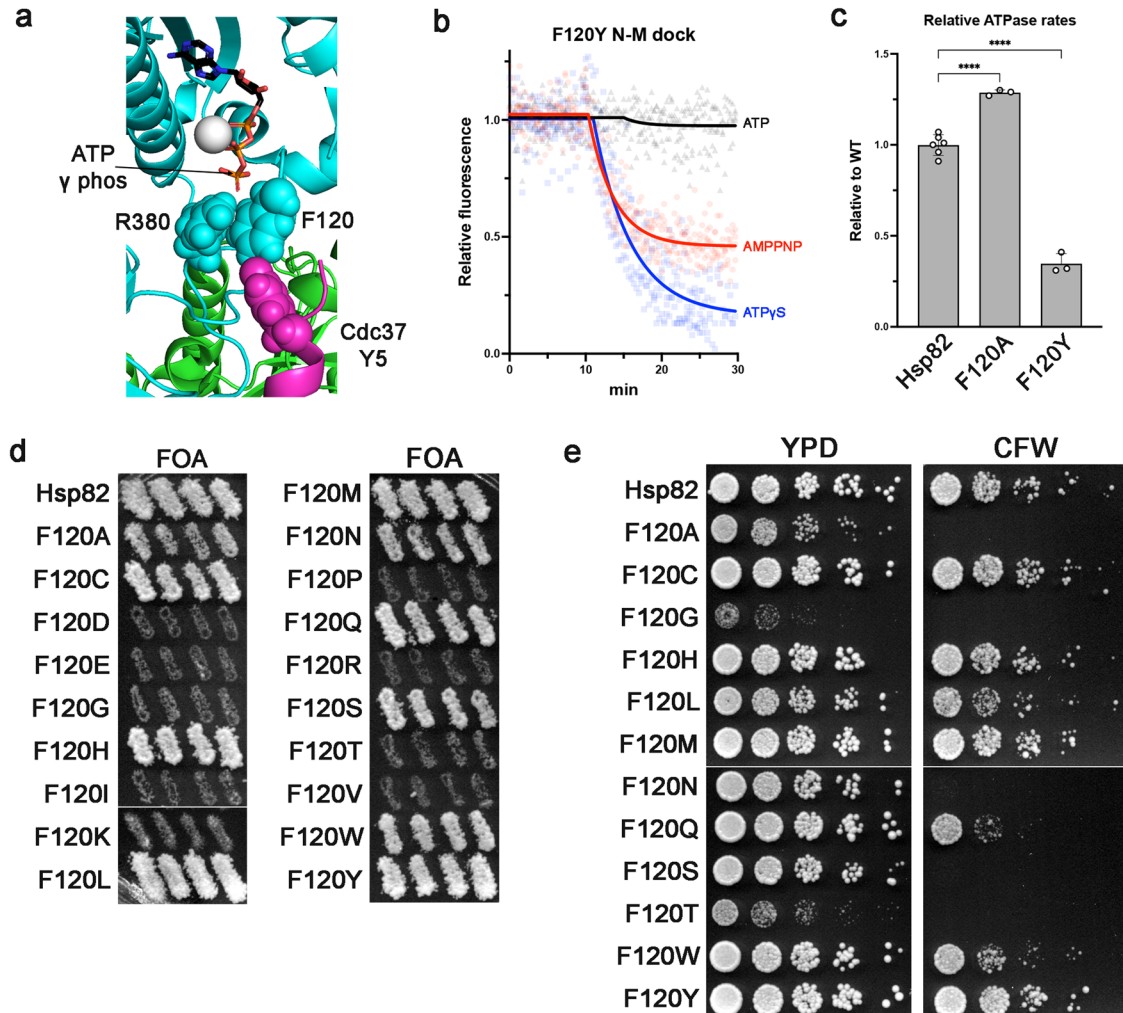

**Fig. 5 | F120 regulates Hsp90 conformational dynamics by properly positioning R380. a** Portion of the Hsp90/Cdc37/Cdk complex crystal structure (pdb: 5FWK) showing the relative positions of ATP (sticks, colored by atom), Hsp90 (cyan) residues R380 and F120 (spheres, Hsp82 numbering) and Cdc37 (magenta) Y5 (spheres). The Mg$^{2+}$ ion is gray and the other Hsp90 protomer is green. **b** Response of Hsp82$^{F120Y}$ to AMPPNP (red), ATP-γ-S (blue) and ATP (black) in the PET assay. Compare to Hsp82 wild type (Fig. 1b) and Hsp82$^{F120A}$ (Fig. 2d). **c** Relative ATPase rates of Hsp82$^{F120A}$ and Hsp82$^{F120Y}$ proteins. Bars are the average rates of three reactions (each replicate is shown as a circle) and error bars are the standard deviation. Asterisks indicate significant difference from wild type ($p < 0.0001$ by one-way ANOVA). **d** Hsp90 functional assay testing all substitutions at position F120. Substitutions F120D, E, I, K, P, R, or V were lethal. Growth of cells expressing Hsp82$^{F120G}$ or Hsp82$^{F120T}$ was apparent only after extended incubation on FOA. Images were cropped from different areas of the same plate or different plates of identical composition. **e** Relative growth (YPD, left) and CFW sensitivity (right) of indicated viable F120 mutants recovered from FOA. Lines in images indicate where portions were cropped from different plates of identical composition.

investigate the role of F120 in conformation dynamics, we introduced F120Y into Hsp82, which is a much more conservative substitution than F120A. We found that Hsp82$^{F120Y}$ formed the closed clamp in response to the three nucleotides very similarly to wild type (Fig. 5b, compare to Fig. 1b), supporting the idea that F120 sterically influences the position of R380 and thus interaction with the γ-phosphate of ATP.

While the F120A mutation slightly increased ATPase compared to wild type, F120Y reduced ATP hydrolysis to about half of wild type (Fig. 5c), again showing that ATP hydrolysis does not correlate with slow growth and CFW$^S$ phenotypes.

We reasoned that if aromatic side chains at position 120 (phenylalanine and tyrosine) functioned well in vivo and smaller residues (alanine) functioned poorly because of the need for bulk to restrict the position of R380, then other bulky residues at position 120 would allow Hsp90 to function while residues with smaller sidechains would not function well or at all. Furthermore, due to the electrostatic interaction between R380 and the γ-phosphate of ATP that is crucial for closed clamp formation, and thus function, we predicted that charged

residues at position 120 would disrupt the γ-phosphate-R380 interaction and be lethal. To test these predictions, we made every substitution at position 120 and assessed the mutants' ability to function in vivo. In line with the latter prediction, yeast cells transformed with plasmids expressing Hsp82 proteins with charged substitutions F120D, E, K or R failed to grow on FOA (Fig. 5d). Substitutions F120I, P, and V were also lethal. Hsp82 proteins with all other amino acids at position 120 functioned in vivo to varying degrees (Fig. 5d, e). Cells expressing Hsp82$^{F120A}$, Hsp82$^{F120N}$, or Hsp82$^{F120S}$ grew slowly (Fig. 5e) while those expressing Hsp82$^{F120G}$ or Hsp82$^{F120T}$ grew so poorly that extended incubation of the FOA plates was needed to recover them. These five slow growing mutants were also CFW sensitive (Fig. 5e). Cells expressing Hsp82 substitutions F120C, H, L, M, Q, W and Y all grew like wild type on rich medium and CFW (Fig. 5e). Thus, amino acids with aromatic sidechains (F, W, Y), bulky sidechains (H, L, M, Q) and cysteine were preferred at position 120, while charged amino acids (D, E, K, R), aliphatic residues that are branched at the β carbon (I, V) and proline were not tolerated at position 120. Substitutions of F120 with amino

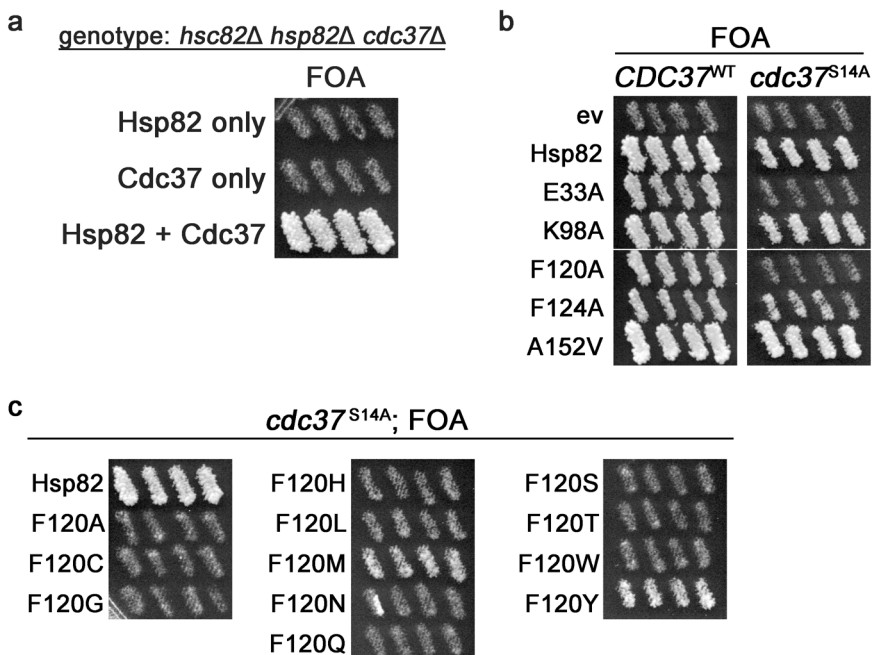

**Fig. 6 | Cdc37 coordinates Hsp90 conformational dynamics through F120.**
**a** Triple knockout strain MR1154 (relevant genotype shown) can only grow on FOA if a viable combination of Hsp90 and Cdc37 alleles are introduced. Cells transformed with Hsp82 only or Cdc37 only (top two rows) failed to grow on FOA while the same strain co-transformed with both Hsp82 and Cdc37 plasmids (bottom row) grew on FOA. **b** Functional Hsp90/Cdc37 assay with the indicated Hsp82 mutants and either wild type *CDC37* (left column) or *cdc37*[S14A] (right column). Cells expressing Cdc37[S14A] and either Hsp82[E33A] or Hsp82[F120A] were inviable. The images in each panel were cropped from the same plate. **c** Like in (**b**) except the viable F120 substitutions in Hsp82 were combined with *cdc37*[S14A]. Only the wild type phenylalanine at position 120 supported normal growth in cells expressing Cdc37[S14A]. The images in (**a**, **c**) were all cropped from the same plate.

acids with short (A, G) and/or polar (N, S, T) sidechains resulted in poor growth and CFW sensitivity. These results are in line with our predictions and support our hypothesis that F120 sterically influences the position of R380 and thus its ability to interact with the γ-phosphate of ATP.

### F120 coordinates a functional interaction with Cdc37 in vivo

F120A was shown to impair activation of heterologously expressed human v-SRC kinase while having little effect on activation of glucocorticoid receptor[53]. The proximity of F120 to the Hsp90-Cdc37 interface and specificity in client maturation defects by F120A suggested to the authors of that study that F120 is important for interaction with the kinase-specific co-chaperone Cdc37[53,54]. Structural studies support this idea and provide insight into the molecular basis for the genetic interaction between F120 and Cdc37[8,55–57]. In cryo-EM structures of human Hsp90 in complex with Cdc37 and various kinase clients, the phenylalanine analogous to yeast Hsp82 F120 is very near to both R380 and the N-terminus of Cdc37[8,55–57] (Fig. 5a). Specifically, F120 and R380 are close to a conserved tyrosine (Y5) in Cdc37 that when mutated to alanine reduces formation of chaperone-kinase complexes[58]. We reasoned that due to its proximity to R380 and Cdc37, F120 may play an important role in coordinating a functional interaction with Cdc37.

First, we asked if any of the mutants in our panel had a genetic interaction with the hypomorphic *cdc37*[S14A] allele, which lacks a phosphorylation site that is important for stabilizing interactions with kinase clients[59,60]. To study genetic interactions between Hsp90 and Cdc37, we constructed a triple knockout yeast strain (*hsp82*Δ, *hsc82*Δ and *cdc37*Δ) that has both *HSC82* and *CDC37* on the same parental *URA3* plasmid (See "Methods"). To grow on FOA, this strain, called MR1154, must be transformed with two differently marked plasmids (e.g. *TRP1* and *HIS3*) that together comprise a viable combination of Hsp90 and Cdc37 alleles (Fig. 6a).

Using strain MR1154 we tested the genetic interactions of the functional Hsp82 mutants with *cdc37*[S14A] and observed that cells expressing Cdc37[S14A] and either Hsp82[E33A] or Hsp82[F120A] were inviable (Fig. 6b). F124A, which like E33A and F120A causes slow growth and CFW sensitivity in the presence of wild type Cdc37 (Fig. 2c), did not exhibit a synthetic growth defect with Cdc37[S14A] (Fig. 6b). This observation suggests that the Hsp82[E33A]-Cdc37[S14A] and Hsp82[F120A]-Cdc37[S14A] phenotypes are not simply an artifact of slow growth and/or CFW sensitivity in the wild type Cdc37 background. Taken together, these results suggest that ATP hydrolysis becomes essential when Cdc37 function is impaired and support the idea outlined above that F120 is important for coordinating Hsp90 conformational dynamics with a functional Cdc37 interaction.

We next tested whether any of the F120 substitutions that were viable in the context of wild type Cdc37 were lethal when combined with *cdc37*[S14A]. We found that only the wild type phenylalanine at position 120 of Hsp82 was able to support robust growth of *cdc37*[S14A] cells (Fig. 6c). Cells expressing both Cdc37[S14A] and either Hsp82[F120M] or Hsp82[F120Y] grew but slowly. The other Hsp82 F120 substitutions that grew normally and were not CFW sensitive in the wild type Cdc37 background (Fig. 5e) were lethal when combined with *cdc37*[S14A] (Fig. 6c). Our results taken together suggest F120 is important for both the γ-phosphate-R380 interaction and functional coordination with Cdc37. Furthermore, the genetic interaction between Hsp82[E33A] and *cdc37*[S14A] suggests Cdc37 also influences hydrolysis and therefore Hsp90 dimer reopening. Thus, Cdc37 has roles in regulating both closing and reopening of Hsp90.

## Discussion

Here we have sought to unravel a persistent mystery: the role of ATP binding in Hsp90 function. The dispensability of Hsp90's ATP hydrolysis means that energy released from hydrolysis is not required for Hsp90's functions in vivo[29,30], but since ATP binding by Hsp90 is

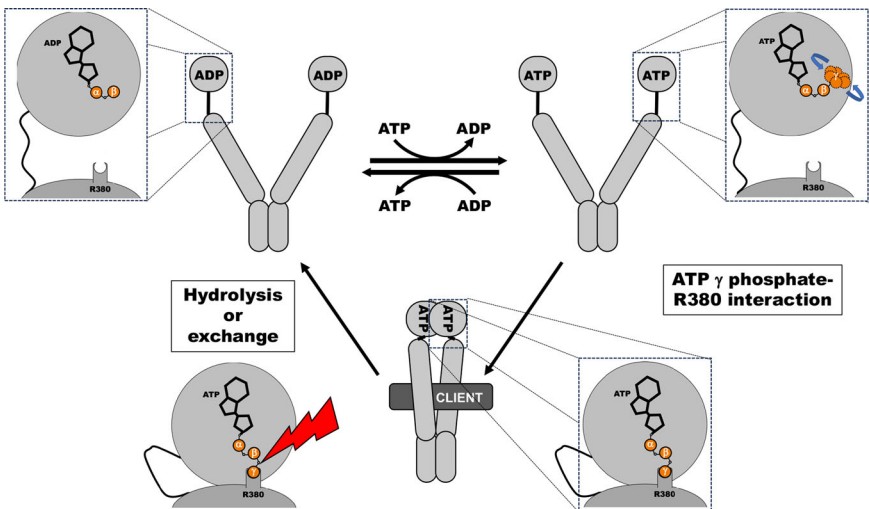

**Fig. 7 | A structural role for ATP in Hsp90 function.** Hsp90 bound to ADP is in the open conformation (upper left, see inset). Exchange of ADP for ATP per se does not induce closing as the γ phosphate of ATP is highly mobile and not interacting with R380 in the M domain (upper right, see inset). Reorientation of ATP within the pocket, which is influenced by factors such as the client and co-chaperones (not depicted for simplicity), brings the γ phosphate of ATP close to R380 so that they can interact and stabilize the closed clamp (bottom, see inset). Hydrolysis of ATP (or exchange to ADP) breaks the connection holding the N and M domain together. ADP in the binding pocket reopens the Hsp90 dimer (upper left).

required, then ATP must be doing something other than providing energy. Our investigations have revealed that ATP itself plays a structural role by stabilizing the closed clamp conformation through its interaction with R380. In our proposed model, formation and dissociation of the closed clamp are essential Hsp90 processes (Fig. 7). ATP facilitates and stabilizes formation of the clamp by physically tethering the N and M domains. Hydrolysis (or exchange) of ATP to ADP destabilizes the closed clamp by severing (or removing) the tether and thus promotes reopening of the Hsp90 dimer (Fig. 7). Thus, Hsp90 needs ATP to close. This idea differs from the prevailing model that posits Hsp90 closes so it can hydrolyze ATP, but is line with conclusions from recent reports that propose ATP hydrolysis serves primarily to reopen closed Hsp90[14,30].

The results of our PET experiments strongly support the idea that bound ATP must orient to the proper conformation for the γ-phosphate to contact R380. Doing so enables ATP to help hold the N and M domains together. The γ-imidophosphate and γ-thiophosphate of the non-hydrolysable analogs AMPPNP and ATP-γ-S, respectively, can "find" R380 more easily than the γ-phosphate of ATP, explaining the differences the individual nucleotides have on Hsp90 conformation that has been observed in many studies. In line with this idea are our findings that altering the three-dimensional topography of the nucleotide binding pocket changes the relative ability of the nucleotides to stabilize the closed clamp. The observation that E372K rescued defects caused by R32A and E33A, but not by mutations in different areas of the pocket, strongly supports this model. Additionally, the reorientation of ATP is influenced by co-chaperones. Sti1 destabilizes the closed clamp in the absence of nucleotide, and notably this effect is stronger when Hsp90 is bound to ATP. Sba1 has the opposite effect as Sti1 and stabilizes the closed clamp in response to ATP. At the N-M interface, F120 helps to properly position R380 to contact the γ-phosphate, through steric effects and possibly through π-cation interactions. This finding together with the genetic interaction between F120 and Cdc37 we described strongly suggest that Cdc37 influences Hsp90's conformational dynamics during kinase client interactions by limiting or enhancing F120's influence on R380's position. Thus, the proper alignment of the γ-phosphate and R380 in vivo may be a rate-limiting event regulated by co-chaperones and other factors such as, but not limited to, the client or posttranslational modifications.

The ability to close is not sufficient to function in vivo, since the nonfunctional mutants Hsp82[G118A] and Hsp82[G123A] retained the ability to close in the PET assay, unlike the other nonfunctional mutants. Perhaps related to this observation was the separate finding that of all the mutants in this study only Hsp82[G118A] and Hsp82[G123A] had a dominant negative growth phenotype. A recent study from the Bolon lab concluded that dominant growth defects caused by mutations in the Hsp82 N domain, including G123H, decoupled ATP hydrolysis and closed clamp formation, resulting in rapid degradation of Hsp90 clients[61]. It is likely that Hsp82[G118A] and Hsp82[G123A] have similar defects as those described in that study. Taken together these observations are in line with our model and demonstrate that proper coordination of Hsp90's conformational changes with hydrolysis are crucial for Hsp90 in vivo function.

After forming the closed clamp Hsp90 must reopen. According to our proposed model (Fig. 7), the simplest way to reopen is to hydrolyze the ATP that helps hold the N and M domains together. However, reopening is not accomplished exclusively by hydrolysis, as it is not required for Hsp90 function in vivo. Exchange for ADP is likely the mechanism by which reopening can occur without hydrolysis, as we recently showed[30]. Another intriguing, yet speculative, mechanism of hydrolysis-independent Hsp90 reopening could be through modulation of the relative position of R380 that disrupts interaction with ATP's γ-phosphate in the closed clamp. These and related lines of inquiry are the focus of current investigations.

The genetic interactions between F120 and Cdc37, together with F120's role in modulating conformational dynamics, lend experimental support to a mechanism that involves functional input from Cdc37 for kinase clients. Perhaps interactions among Cdc37's N-terminus, R380, and F120 provide a structural basis for a role in Cdc37 in regulating both Hsp90's closing and reopening. Furthermore, the synthetic growth defect due to combining the hypomorphic *cdc37*[S14A] and the hydrolysis defective *hsp82*[E33A] alleles shows that when Cdc37 function is compromised, Hsp90's ATP hydrolysis becomes essential. This result further highlights the important role Cdc37 plays in Hsp90 function and presents a testable hypothesis as to why Cdc37 is essential, unlike most other Hsp90 co-chaperones.

Some Hsp82 mutant proteins that did not form the closed clamp or support viability retained the ability to hydrolyze ATP, an observation reported earlier for Hsp82[R32A] and Hsp82[R380A][13,29,47]. Although the

non-closing mutant proteins all had ATPase rates lower than wild type Hsp82, these and other findings demonstrate that ATP hydrolysis can occur in the open state and is strong evidence that argues against the core idea of the current model that hydrolysis only occurs in the closed clamp. Thus, the use of ATP hydrolysis as a proxy for closed clamp formation seems problematic under some experimental circumstances.

Much further experimentation is needed to fully understand the regulation of ATP hydrolysis in the context of ATP being a structural component of the Hsp90 closed clamp rather than simply a source of energy for refolding clients. Deeper investigations into such a non-traditional role for ATP may foster a more meaningful understanding of Hsp90's essential functions, and its place within the molecular chaperone family. Additionally, identifying chemical compounds that target or stabilize specific conformational states of Hsp90 rather than simply block ATP binding may illuminate new avenues of Hsp90-based therapies.

The interaction of the M domain residue R380 with the γ-phosphate of ATP in the N domain strongly resembles the widely conserved "arginine finger" found in other NTPases. For example, multimeric AAA+ ATPases have an arginine that protrudes into the nucleotide binding pocket of a neighboring subunit and interacts with the γ-phosphate of ATP bound there[62–64]. Many of these interactions are important for both ATP hydrolysis and oligomer formation, but an arginine to alanine substitution in the membrane fusion factor NSF reduces its activity by disrupting subunit interactions without impairing ATP hydrolysis[65]. Similarly, arginine fingers stabilize plant transport complex GTPases and arginine to alanine replacements disrupt their ability to form dimers, yet only modestly affect ATP hydrolysis[66]. Septins also rely on such interactions to form complexes, with GTP-arginine binding mediating interactions between subunits of septin complexes. In some septins the arginine is replaced by a histidine that contacts the γ-phosphate of GTP in a partner septin without activating GTP hydrolysis, which evolved to restrict subunit-subunit interactions to specify arrangements of septins in complexes[67,68].

These observations align with a view that a general role of NTP-arginine finger interactions is to promote conformational rearrangements, separate from or linked to hydrolysis, that are needed for the biological activity of the complexes. Although in these examples it is implied that ATP or GTP is needed to mediate subunit interactions, which could be separable from hydrolysis, the focus has been on the trans-acting amino acid, and we have not found a study that explicitly hypothesizes or concludes that the nucleotide itself plays a structural role, or directly tests this idea experimentally, as we do here.

Viewing ATP as a severable structural component of the arginine finger motif may lead to new insights regarding AAA+ ATPase or other arginine finger-containing proteins' mechanisms and resolve questions surrounding the roles of nucleotide binding and hydrolysis in those systems. Due to the similarities between R380 and arginine fin-gers we suggest that R380 represents a class of arginine finger that acts to stabilize inter-domain interactions, as proposed earlier[13], rather than inter-subunit interactions.

Regarding Hsp90, ATP may be likened to an "intramolecular Velcro", a structural cofactor that stabilizes the interaction of the N and M domains of Hsp90 in a reversible manner. Small molecule stabilizers of protein-protein interactions, so-called "molecular glues", have been described in nature[69]. For example, inositol polyphosphates stabilize the interaction of histone deacetylases (HDACs) and adapter proteins and are essential for HDAC enzymatic function[70]. However, to our knowledge, a role for ATP such as what we are proposing has not been described.

## Methods

### Yeast media, strains and plasmids

Nonselective *S. cerevisiae* growth medium was YPD (1% yeast extract, 2% peptone, 2% dextrose). Liquid cultures of parental, non-transformed strains that were Trp⁻ were grown in YPD supplemented with 0.01% tryptophan. Selective medium was synthetic complete (SC, 2% glucose, 0.67% yeast nitrogen base with ammonium sulfate, SC drop-out mix as appropriate (Sunrise Scientific)). Solid media contained 2% agar. Diploid strains were sporulated on 2% agar supplemented with 1% potassium acetate. When used, calcofluor white (Sigma F3397) was added to YPD plates at 5 mg/L. 5'-fluoro-orotic-acid (FOA, US Biological F5050) was used in SC dropout plates at a final concentration of 1 g/L. All cells were grown at 30 °C unless indicated otherwise. Yeast strains used in this study are listed in Table 2. All strains are isogenic to BY4741[71]. Construction of strains MR1075, MR1143, MR1079, MR1083 and MR1081 is described elsewhere[37]. Strain MR1155, used to check the expression of inviable Hsp82 proteins (see below), is isogenic to MR1075 except it has plasmid pMR520 instead of pMR62. It was constructed using techniques described previously[37]. Strain MR1110 is isogenic to MR1075 except it has knockouts of both *CPR6* and *CPR7* and was made by standard genetic techniques.

To construct strain MR1154, first the heterozygous diploid knockout strain BY4743 *cdc37*ΔKanMX⁺/⁻ (Horizon YSC6274-201926094) was transformed with plasmid pMR411 (See Table 3). After sporulation and tetrad dissection, a G418 resistant, Ura⁺, FOA sensitive clone with mating type a was crossed to strain MR1060[37]. This diploid strain (heterozygous for *HSC82*, *HSP82*, *CDC37* and *TRP1*) was grown on nonselective medium to allow loss of both *URA3* plasmids from the parents (pMR62 and pMR411), then grown on FOA to give strain MR1153. MR1153 was transformed with plasmid pMR542, sporulated and dissected. A spore clone that was resistant to hygromycin, nourseothricin and G418, FOA sensitive and Trp⁻ was selected and named MR1154.

Plasmids used in this study are listed in Table 3. Construction of plasmids pMR62, pMR325, pSK59, and pMR363 have been

## Table 2 | *Saccharomyces cerevisiae* strains used in this study

| Strain | Genotype | Ref |
|---|---|---|
| MR1075 | *his3Δ1; leu2Δ0; lys2Δ0; trp1Δ63; ura3Δ0; hsc82ΔHphMX; hsp82ΔNatMX; pMR62 (P_HSC82::HSP82/URA3)* | 37 |
| MR1155 | *his3Δ1; leu2Δ0; lys2Δ0; trp1Δ63; ura3Δ0; hsc82ΔHphMX; hsp82ΔNatMX; pMR520 (HSC82/URA3)* | This study |
| MR1143 | *his3Δ1; leu2Δ0; lys2Δ0; trp1Δ63; ura3Δ0; hsc82ΔHphMX; hsp82ΔNatMX; sti1ΔKanMX; pMR62 (P_HSC82::HSP82/URA3)* | 37 |
| MR1079 | *his3Δ1; leu2Δ0; lys2Δ0; trp1Δ63; ura3Δ0; hsc82ΔHphMX; hsp82ΔNatMX; aha1ΔKanMX; pMR62 (P_HSC82::HSP82/URA3)* | 37 |
| MR1110 | *his3Δ1; leu2Δ0; trp1Δ63; ura3Δ0; hsc82ΔHphMX; hsp82ΔNatMX; cpr6ΔKanMX; cpr7ΔKanMX; pMR62 (P_HSC82::HSP82/URA3)* | This study |
| MR1083 | *his3Δ1; leu2Δ0; lys2Δ0; trp1Δ63; ura3Δ0; hsc82ΔHphMX; hsp82ΔNatMX; ppt1ΔKanMX; pMR62 (P_HSC82::HSP82/URA3)* | 37 |
| MR1081 | *his3Δ1; leu2Δ0; lys2Δ0; trp1Δ63; ura3Δ0; hsc82ΔHphMX; hsp82ΔNatMX; sba1ΔKanMX; pMR62 (P_HSC82::HSP82/URA3)* | 37 |
| MR1153 | Diploid; *his3Δ1⁺/⁺; leu2Δ0⁺/⁺; lys2Δ0⁺/⁻; trp1Δ63⁺/⁻; ura3Δ0⁺/⁻; hsc82ΔHphMX⁺/⁻; hsp82ΔNatMX⁺/⁻; cdc37ΔKanMX⁺/⁻;* | This study |
| MR1154 | *his3Δ1; leu2Δ0; lys2Δ0; trp1Δ63; ura3Δ0; hsc82ΔHphMX; hsp82ΔNatMX; cdc37ΔKanMX; pMR542 (HSC82, CDC37/URA3)* | This study |

**Table 3 | Plasmids used in this study**

| Plasmid | Backbone | Insert | Ref |
|---|---|---|---|
| pMR62 | pRS316 | P$_{HSC82}$::*HSP82* ORF | 72 |
| pMR325 | p414-GPD | *HSP82* ORF | 37,77 |
| pMR411 | pRS316 | *CDC37* -/ + 500 bp flanking | This study; 74 |
| pMR411H | pRS313 | *CDC37* -/ + 500 bp flanking | This study; 74 |
| pMR520 | YCpLac33 | *HSC82* -/ + 500 bp flanking | This study; 75 |
| pMR542 | YCpLac33 | *CDC37* -/ + 500 and *HSC82* -/ + 500 | This study |
| pSK59 | pET28b | *HSP82* ORF | 73 |
| pSK91 | pET28b | *STI1* ORF | This study |
| pMR363 | pET28b | *SBA1* ORF | 73 |

described[37,72,73]. Plasmid pMR411 contains the *CDC37* open reading frame flanked by 500 bp of its native promoter and terminator sequences on a *Bam*HI fragment cloned into pRS316[74]. pMR411H is identical except the same fragment was cloned into pRS313[74]. pMR520 contains the *HSC82* gene with 500 bp of native promoter and terminator sequences on a *Bam*HI fragment. It was made by subcloning the *Bam*HI fragment containing *HSC82* from plasmid pMR55W[30] into YCpLac33[75]. Plasmid pMR542 contains both *HSC82* and *CDC37* genes (the two genes are on opposite strands and expressed from their native promoters). It was made by first amplifying the *CDC37* gene (using pMR411H as template) to add appropriate homology (20 bp) to *Sac*I-digested pMR520. The insert was then cloned into *Sac*I-digested pMR520 using NEBuilder (NEB E2621S). pSK91 contains the Sti1 open reading frame amplified to add *Spe*I and *Xho*I sites (using yeast genomic DNA as template) and cloned into the *Nhe*I and *Xho*I sites of pET28b. Mutations were introduced into plasmids using the Quick-Change Lightning Multi site directed mutagenesis kit (Agilent 210515). Primers used in this study are listed in Supplementary Table 1.

### Protein Purification

Purification of His-tagged Hsp82 proteins was performed using Talon (TakaraBio 635503) affinity chromatography followed by DEAE Sepharose Fast Flow (GE Healthcare 17-0709-01) and/or gel filtration on Superdex 200 Increase 10/300 GL (Cytiva 28-9909-44), as described[30,37]. Proteins used in the PET assays (see below) contained additional substitutions E192C and N298W[11].

Purification of Hsp90 cochaperones Sti1 and Sba1 was performed using Talon affinity chromatography as described[37,73]. The His-tag was removed by thrombin digestion and reapplication over the Talon column.

### PET measurements

PET measurements were performed essentially as described[30,37]. Proteins containing the E192C and N298W substitutions (in addition to other mutations as indicated) were labeled with 1.3-fold molar excess of Atto Oxa-11 maleimide (AttoTec AD Oxa11-41) for 2 hr at room temperature. Labeling was done either after the DEAE step and prior to gel filtration (to remove unreacted dye) or after purification and then desalted to remove unreacted dye. 2 µM (monomer) of unlabeled proteins (that is, not containing E192C and N298W) was mixed with 0.1 µM labeled proteins in 100 µL reactions. When present, co-chaperones were 2 µM, and trimethylamine N-oxide (TMAO, Sigma T0514) was 1 M, and were preincubated with Hsp82 for -10 min prior to T = 0. Fluorescence (620 nm excitation/680 nm emission) was measured every 20 s for 10 min using a BMG Omega plate reader before injection of nucleotide, 2 mM final for AMPPNP (Roche 10102547001), ATP-γ-S (Sigma A2383), or ATP (Roche 11140965001) as indicated, then measured every 20 s for 20 additional minutes. All data points

(analyzed using Omega MARS version 3.32 R2, Microsoft Excel 16.97.2 and GraphPad Prism 10.2.2) from three or four independent experiments are shown and lines are the exponential decay fits (calculated using Prism 10).

### ATPase

ATPase rates were measured using the pyruvate kinase/lactate dehydrogenase (PK/LDH, Sigma P0294) coupled assay as described[30,37], in 100 µL volumes with 2 µM (monomer) Hsp82 proteins and 2 mM ATP. The ATPase rates were subtracted by those of identical reactions containing 100 µM radicicol, to correct for contaminating ATPases. Data presented are from three independent replicate reactions and are relative to wild type Hsp82, and were analyzed using Omega MARS version 3.32 R2, Microsoft Excel 16.97.2 and GraphPad Prism 10.2.2. Bars are the average values and the error bars are the standard deviation. Statistics (one-way ANOVA compared to wild type) were calculated using Prism 10.

### Plasmid shuffling/Hsp90 in vivo functional assay

We tested Hsp90's essential in vivo functions by FOA counterselection, as described[30,37,73]. Yeast strains that had deletions of both chromosomal Hsp90 genes (and other mutations as indicated, see Table 2) harbored a *URA3*-marked plasmid containing *HSP82* (pMR62) to provide Hsp90 function. Cells were transformed with *TRP1*-marked plasmids encoding Hsp82 variants. Four independent transformants were isolated and patched onto a master plate containing uracil, to allow loss of plasmid pMR62. Following overnight growth at 30 °C, the master plate was replica-plated to identical medium containing FOA, incubated at 30 °C for two days, then scanned. Growth on FOA indicated a functional Hsp90 allele encoded by the *TRP1* plasmid which allowed loss of plasmid pMR62. Cells that were transformed with an inviable Hsp90 variant could not lose the parental pMR62 plasmid and were killed by FOA, which is toxic to cells expressing Ura3 protein. Rarely, papillae arose from single isolates on FOA plates (for example: Fig. 3a, G123A and Fig. 6c, F120N). These FOA resistant cells were usually the product of plasmid-plasmid recombination (that is, the wild type Hsp82 transferred from the parental *URA3* plasmid to the *TRP1* plasmid) or mutations in *URA3* that allowed the parental wild type plasmid to escape FOA toxicity. Such events are rare but inadvertently selected for under these conditions, and therefore these papillae are technical problems and not true complementation. Therefore, viability and thus Hsp82 function was scored as confluent growth of all four biological replicates on FOA.

### Spot assays

FOA-resistant cells from plasmid shuffle experiments were used to inoculate liquid YPD medium for overnight growth. The OD$_{600}$ of the cultures were normalized to 0.25 via dilution with water, serially diluted 1:5 four times, and 4 µL of each dilution was spotted onto solid media as indicated. Plates were scanned after 2 days incubation at 30 °C.

### Growth Curves

To measure growth rates of cells expressing the indicated Hsp82 mutant only, FOA-resistant cells (three independent biological isolates of each variant) were isolated and grown in YPD. For assessment of dominant negative growth effects, isolates were picked from primary transformation plates and grown in SC medium lacking tryptophan. After overnight growth, cells were diluted 1:100 (or appropriate to attain an initial OD$_{600}$ of 0.05–0.1) in 150 µL of fresh medium in a 96 well plate. Absorbance at 600 nm was measured every 20 min for 18 hr, with shaking (200 rpm), at 30 °C, in a BMG Omega plate reader. Data were analyzed using Omega MARS version 3.32 R2, Microsoft Excel 16.97.2 and GraphPad Prism 10. Growth rates were calculated from the steepest linear 5 hr window (R$^2$ ≥ 0.99 when plotting the raw data on a

log scale) using the following equation: doubling time $= \frac{5 * \ln 2}{\ln \frac{end \, A600}{start \, A600}}$. The data are presented as cell divisions per hour, that is, the inverse of the doubling time. Bars are the average of the replicates and error bars are the standard deviation. Statistics (one-way ANOVA compared to wild type) were calculated using Prism 10.

## Western blotting

Analysis of expression of the inviable Hsp82 variants was done using an antibody (Enzo ADI-SPA-840) that recognizes the Hsp82 isoform of yeast Hsp90 but not Hsc82. Plasmids encoding the indicated Hsp82 variants, or an empty vector control, were used to transform strain MR1155 (see Table 2), which harbors a *URA3* plasmid encoding *HSC82*. Transformants were grown overnight in liquid SC -Trp media. 50 $OD_{600}$ units of cells were harvested by centrifugation, washed with ice cold water supplemented with 2 mM PMSF, pelleted and stored at −80 °C. Cells were lysed by resuspending pellets in 500 μL of TBS containing 0.1% Triton X-100 and protease inhibitors, adding an equal amount of glass beads, and processing $2 \times 45\,s$ in a bead beater. Lysates were clarified via centrifugation (5 min at $10,000 \times g$), and a portion was mixed 1:1 with 2x SDS loading dye. 5 μL of this mixture was loaded onto 4–20% Criterion TGX (Bio-Rad 5671095) gels, separated by electrophoresis and transferred to PVDF membranes. Duplicate membranes were probed with anti-human Hsp90α (Enzo Life Sciences ADI-SPA-840, clone 9D2, which recognizes Hsp82 but not Hsc82) or anti-Hsc82 (AbCam ab30920, polyclonal, which recognizes both Hsp82 and Hsc82). Equal loading was verified by staining the membranes with amido black after blotting and image capture.

## Reporting summary

Further information on research design is available in the Nature Portfolio Reporting Summary linked to this article.

## Data availability

The data generated in this study are provided in the Supplementary Information/Source data file. Figure 5a contains a portion of PDB 5FWK. Source data are provided with this paper.

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

## Acknowledgements
We thank our NIH colleagues for helpful discussions and critical reading of the manuscript. This research was supported by the Intramural Research Program of the NIH, the National Institute of Diabetes and Digestive and Kidney Diseases (NIDDK), award number ZIA DK024946-24 (D.C.M.).

## Author contributions
M.R. conceived and performed experiments and wrote the manuscript. D.C.M. conceived experiments and assisted writing the manuscript.

## Funding

## Competing interests
The authors declare no competing interests.
