## [Transparent Peer Review file · Nature Communications]

ATP plays a structural role in Hsp90 function

Corresponding Author: Dr Michael Reidy

Version 0:

Reviewer comments:

Reviewer #1

(Remarks to the Author)

583787_0_art_file_10301417_sr93qb

This manuscript by Reidy and Masison represents a very thorough analysis of the role of ATP binding in Hsp90 function, combining in vitro biochemistry with in vivo genetics. The extent to which the conformational changes driven by ATP hydrolysis contribute to Hsp90 function has been a long-standing unresolved issue, and here the authors rely on a pair of powerful systems further empowered by detailed structural insights to ask incisive questions. I found the data to be very compelling and, considering the complexity of the ideas involved, very clearly presented. I think this work will be a valuable contribution to the field, and will be of interest to the broad readership of the journal. That said, I have some suggestions for improving the readability of the manuscript, and one suggestion for an easy experiment that could be informative.

Suggested experiment: The addition of TMAO to the PET assay was informative. TMAO can also be added to yeast growth media and can affect protein conformations in vivo. It would be interesting to test the viability of some of the mutants in the presence of TMAO.

Suggested revisions:

Discussion: The proposed structural role of the gamma phosphate of the nucleotide triphosphate reminds me of what has been proposed for the role of gamma phosphate of GTP in mediating subunit-subunit interactions within septin hetero-oligomeric complexes, where certain septins have independently evolved to become "pseudoGTPases" to restrict the kinds of subunit-subunit interactions that occur in vivo (see PMID 31990274, for example). Indeed, it seems that an Arg finger that is still present in some septins that are active GTPases was replaced during evolution by a His that contacts the gamma phosphate of a partner septin without triggering GTP hydrolysis (see PMID 38994454). If the authors also think this is a useful comparison, it may be worth mentioning.

Abstract: Given the very broad readership of Nature Communications, it would be worth providing a tiny bit more context to the beginning of the paper, such as stating that Hsp90 is a highly conserved molecular chaperone that binds client proteins via a "clamp" to promote their native folding.

The third sentence of the Abstract is also worded awkwardly: "mutating the conserved arginine R380, which interacts with the gamma phosphate of ATP in Hsp90's closed clamp conformation, and other residues". It is difficult to know if the "other residues" is the object of "mutating" or of "interacts with". Please re-word.

Later in the Abstract: "ATP must properly align to interact with R380". In the current wording, it is unclear with what ATP must properly align; is it something else, or is it R380?

The final sentence of the Abstract: "This in turns suggests that some arginine fingers might be structural elements important for regulation of inter-subunit or -domain interactions...". The current wording could be construed as implying that this manuscript is making a novel suggestion that arginine fingers might regulate inter-subunit interactions apart from hydrolysis, when in fact this idea is already present in the literature (see above).

The first use of "yeast" on p. 2 line 15 should be specified as budding yeast or, better, *Saccharomyces cerevisiae*, since Hsp90 in fission yeast (for example) is not called Hsp82 and the Arg residue in question is R375, not R380.

p. 4 Line 17: "This idea is supported by the observation that the gamma-phosphate of ATP is not visible in the crystal

structure of the Hsp90 N domain and is thus mobile". I don't quite follow the logic here. Is the idea that the crystal structure of the Hsp90 N domain is in neither the "open" nor the "closed" conformation, because the M domain is absent and therefore "open" or "closed" cannot be assigned, and the gamma-phosphate is mobile because it is not "locked" into one conformation or the other ("open" or "closed")? I'm missing a logical step here about how the mobility of the gamma-phosphate in the N domain structure is relevant to the orientation that ATP must adopt in order for the clamp to close. Can the authors please spell this out a little better?

p. 5 Line 16-17: "since cells transformed with plasmids encoding this mutation were unable to grow on medium containing 5'-fluoro-orotic acid (FOA)": it might help the reader unfamiliar with this assay to add "URA3-marked" before "plasmids" here. Otherwise, it might seem that somehow FOA acts differently depending on whether Hsp104 is wild-type or mutant per se.

Figure 1i: It would be very helpful here to include a mutant that *is* sensitive to CFW, to demonstrate that the CFW activity/concentration in the medium was sufficient for the intended purpose in this particular experiment.

p. 6 line 6: "were mutated": it seems that "are mutated" would make more sense here.

Similarly, p. 6 line 8: "we chose residues that were conserved": "are conserved" makes more sense, since they will always be conserved.

p. 7 line 16: "or Sba1 or growth" should be "or Sba1 for growth".

p. 8 lines 1-2: "the ability to form the closed clamp – with any nucleotide – is the prerequisite for supporting viability." The "with any nucleotide" part here is a little confusing, since it would seem to refer to ATPgammaS or AMPPNP, but the relevant assay is viability, and ATPgammaS or AMPPNP will never be found in vivo.

p. 13 line 28: "Some inviable Hsp82 mutant proteins": it is a *very* minor point, but it is the mutant cells that are inviable, not the mutant proteins per se, so this should probably be reworded as "Some mutant Hsp82 proteins that did not support cell viability and did not form the closed clamp..." Also, in some places "non-viable" is used instead of "inviable". I don't have a strong preference, but the usage should be consistent throughout, to prevent confusion.

Figure 5d,e and Supplemental Figure 5: it appears that the colonies/patches shown are sometimes from different plates, because there is an obvious "splicing" together of images. This makes sense, given that it would take a very large plate to accommodate so many strains side-by-side, and this is not a problem per se. However, I strongly recommend that the authors insert visible lines to indicate where images have been adjoined, and that in the figure legend, the authors indicate if the colonies/patches shown were grown on the same or on different plates. There may be some plate-to-plate variation (certainly we see this in our hands) and while it is not a major concern, it is best to be transparent about how the experiments were done.

Reviewer identity: Michael McMurray, PhD, University of Colorado School of Medicine

Reviewer #2

(Remarks to the Author)

The manuscript by Reidy and Masison reports on a large set of experiments to investigate the role of ATP in the function of Hsp90. Therefore, they use a combination of photon-induced electron transfer (PET), ATPase assays and various yeast viability studies. As far as I can judge, the experiments are well conducted.

The function of ATP and its hydrolysis in Hsp90 is an important question, which has already caught a lot of attention resulting in dozens of publications. In my view, many of the findings in this manuscript are either not new or largely speculation as detailed in the following.

The main conclusion of the manuscript is that the role of ATP in Hsp90 function is structural, namely that ATP is a severable linker that facilitates both the formation and dissociation of the closed clamp structure of Hsp90.

In my view, this is not much different from one of the conclusion of the following publication (which has obviously escaped the authors' attention – at least it is not cited): Wolf et al., DOI: 10.1039/d0sc06134d . In this publication from 2021 it already has been shown that the hydrolysis of one ATP is coupled to a conformational change of Arg380, which in turn passes structural information via the large M-domain alpha-helix to the whole protein. It even describes the stability of the closed clamp under various nucleotide conditions and the subsequent dissociation of the closed clamp.

Altogether, I appreciate that with this new manuscript more evidence has been found for the structural role of ATP, but the main finding that ATP in Hsp90 has a structural role is not new.

Therefore, altogether, I cannot support publication in Nature Communications.

Major points:

1) Result 1: "Positive charge at position 380 is necessary for closed clamp stabilization and in vivo Hsp90 function". This has been shown before: See references: DOI: 10.1002/pro.2103 and DOI: 10.1039/d0sc06134d.

2) Result 2: "Alterations in the nucleotide binding pocket result in different conformational responses to ATP and ATP analogs." This is also clear from several previous studies, e.g. a very recent one in Nat. Commun. (which also seemed to have escaped the authors' attention): <https://doi.org/10.1038/s41467-024-52995-y>

In addition, I find it troublesome that there is no correlation between the effects the mutations had on the ability of the three nucleotides to stabilize closed clamp formation and phenotype. How can the authors explain that the mutations that support viability sometimes show stabilization with ATP and in other cases with ATP analogs? In other words: Why is only stabilization with any one nucleotide necessary to see viability? In the cell, Hsp90 should only see ATP, therefore

stabilization with ATP should be most important.

3) Result 3: "E372K rescues growth defects of R32A and E33A and restores closed clamp stabilization by ATP-g-S in vitro" – This in vitro result with an ATP analog is new.

4) Result 4: "Co-chaperones influence Hsp90's conformational dynamics differently in response to ATP or AMPPNP". This has been shown before several times, e.g. also in <https://doi.org/10.1038/s41467-024-52995-y>. In addition, here the authors write that AMPPNP is structurally predisposed to induced closed clamp formation, while before the authors' showed that stabilization by other nucleotides (ATP-g-S, ATP) is also sufficient.

5) Result 5: "F120 regulates Hsp90 conformational dynamics by properly positioning R380". The data on the F120 mutation is very interesting, but I do not see any data on conformational dynamics.

6) Result 6: "Cdc37 coordinates Hsp90 conformational dynamics through F120". Again, I do not see any data on conformational dynamics.

Minor comments:

- Title: I do not understand what the authors mean by "non-energetic"? I have not heard that word before. In my view, also the here described switch uses (transforms) energy, therefore it should be energetic.

- P.13, line 2/3: "... Sti1 destabilizes the closed clamp even more when Hsp90 is bound to ATP than in the apo state...".

Before, the authors wrote that there is not much closed clamp of Hsp90 (WT) in the ATP or apo state anyway. How does this fit together?

- Fig. 4: When are Sti1, Sba1 and TMAO added?

- For the PET measurements, the authors write "shown are the representative data sets of at least three independent experiments". Does this mean that only one data set is shown? Or are all data sets are shown? Or is the average of the data sets shown? If only one data set is shown, what is the variation in the decay?

Reviewer #3

(Remarks to the Author)

How exactly ATP binding and hydrolysis regulate the mechanism of Hsp90 has remained an open issue. This study suggests that ATP binding to Hsp90 plays an important structural role in the chaperone mechanism bringing different parts of the Hsp90 molecule together. The authors use an established PET-assay for Hsp82-NM docking together with experiments in yeast to study several mutants also in combination with co-chaperones and a molecular crowder.

The find, that mutation of R380 and mutations close to the nucleotide binding pocket alter the formation of the closed clamp upon addition of ATP, ATPγS and AMP-PMP. This supports the idea that ATP must be present in a specific orientation to drive conformational changes leading to the closed clamp conformation (tethering the N and M domains).

The orientation of ATP is also influenced by co-chaperones: Sti1 destabilizes the closed clamp whereas Sba1 stabilizes the closed clamp in response to ATP. It is suggested that Cdc37 regulates Hsp90 conformational dynamics by limiting or enhancing the influence of F120 on R380. The cellular environment/molecular crowding may shift the equilibrium further to the closed clamp formation.

Taken together, this is an interesting study that sheds new light on the consequences of ATP binding to Hsp90 and its regulation. Additional data reporting on nucleotide binding directly may further strengthen the study.

Specific points

1. ITC experiments may provide additional insight and direct evidence to what extent the mutants used in this study affect the affinity and thermodynamic parameters of nucleotide binding.

2. Figure 2c: The mutant G118A forms a closed clamp with ATP, also the mutant G123A with ATP and ATPγS but both are not viable. Is there an explanation for this?

3. Would the PET-docking experiment with Cdc37 and Cdc37_S14A together with Hsp82, E33A and F120A lead to the same conclusions as proposed based on the in vivo assays in figure 6?

4. A recently published paper on loss-of function and dominant negative mutations has some overlap with the mutants used in this work: Flynn, Julia M., Margot E. Joyce, and Daniel N. A. Bolon. 2025. This study should be included in the discussion

5. Statistical significance should be determined for numerical data shown in the various bar graphs.

Minor

For direct comparison, the ATPγS traces of figures 3c and 2a and 3d and 1b should be combined in one figure.

In figure 1f, "rates" is lower case, whereas in figure 1h it is upper case.

Figures 3 and 4: The font size is much larger than in the other figures.

Figure 5C: for the mutant F120Y only duplicates are shown.

Page 2, line 11: "co-called" - "so-called"

Version 1:

Reviewer comments:

Reviewer #1

(Remarks to the Author)

I am satisfied with the authors' revisions to the manuscript and how they address my and other reviewer concerns.

Reviewer #2

(Remarks to the Author)

The authors have substantially improved their manuscript. They have incorporated new data on mutations, enhanced the integration of their findings within the existing literature, and refined the comparison between *in vivo* and *in vitro* results. As outlined in my initial review and elaborated upon below, I still do not fully agree with all aspects of the data interpretation or the claimed novelty. However, given the central importance of ATP in Hsp90 function, it is essential to allow for multiple perspectives. Therefore, I now support the publication of this manuscript, though I believe the following points should still be addressed:

1. Many researchers, including myself, have been influenced by the extensive body of work on molecular motors such as myosins and kinesins, where ATP binding, hydrolysis, and phosphate release are well-established as structural drivers of the mechanochemical cycle. In that context, it seemed intuitive that each step has a structural function in ATPases. Nevertheless, I agree that explicitly stating this for Hsp90 is important. The revised wording, particularly the removal of the term "non-energetic," represents a clear improvement, but maybe the authors can even further clarify their wording with this information.
2. I acknowledge that the authors' approach to examine these mutations, cochaperones, and different nucleotides in this combination is novel. However, several of these individual components and their interactions have been studied previously. I cannot provide a comprehensive literature review here and I appreciate that this also cannot be done for this manuscript, but for example, the following publication investigated the effects of Sti1 and Sba1 in the context of ATP and AMP-PNP (see DOI: 10.1038/nsmb.1965), i.e. some of the main components in this work.
3. I find the following new sentence problematic: "ATP must induce closing *in vivo* since mutations that block closing or ATP binding are lethal." While this might be true, it should be expressed more cautiously, because strict causality is not given here. For instance, the mutations might inhibit closing through mechanisms independent of ATP, or the inability to bind ATP could impair other critical steps (such as lid closure) rather than directly preventing N-terminal domain closure.

Reviewer #3

(Remarks to the Author)

The authors have answered the queries raised by me in detail.

I appreciate their comments on determining the affinity of ATP and Hsp90 by ITC - and the description of the results of their efforts in using ITC for Hsp90 analysis. It is unfortunate that there seem to be technical issues that preclude the use of ITC in this context.

The authors may nevertheless comment on the effects of mutations on the affinity of ATP binding and how this may or may not affect the results obtained.

The other points raised were answered in a satisfactory manner.

A general comment: I think split ATPases (GHKL ATPases) are in general constructed in a way that ATP binding induces conformational changes and that hydrolysis then resets the system.

Response to reviewers.

We thank the reviewers for their time and expertise in reviewing our manuscript. At the reviewers' requests, we have rewritten a substantial portion of the Abstract to provide more clarity and background, renamed subsections in the Results, reworded confusing sentences, and added citations. We have updated the figures and legends with statistical analyses, image clarification, controls and more replicates at the reviewers' requests. To address a specific concern raised by the reviewers, we have included new data on the dominant negative effects of the two mutants that closed in the PET assay but were unable to support viability (G118A and G123A). To ease presentation of the various correlations in the data points we have also rearranged some data in Figure 2, moved some data to supplemental Figures and reorganized the Supplemental Figures in the revised version. The Results and Discussion was updated in conjunction with these additions and rearrangements as necessary. We also made changes to Table 1, to summarize both the in vivo and in vitro findings in one place to facilitate comparisons.

The reviewer comments are pasted below. Our point-by-point response is in *italics*.

Reviewer #1

This manuscript by Reidy and Masison represents a very thorough analysis of the role of ATP binding in Hsp90 function, combining in vitro biochemistry with in vivo genetics. The extent to which the conformational changes driven by ATP hydrolysis contribute to Hsp90 function has been a long-standing unresolved issue, and here the authors rely on a pair of powerful systems further empowered by detailed structural insights to ask incisive questions. I found the data to be very compelling and, considering the complexity of the ideas involved, very clearly presented. I think this work will be a valuable contribution to the field, and will be of interest to the broad readership of the journal. That said, I have some suggestions for improving the readability of the manuscript, and one suggestion for an easy experiment that could be informative.

Suggested experiment: The addition of TMAO to the PET assay was informative. TMAO can also be added to yeast growth media and can affect protein conformations in vivo. It would be interesting to test the viability of some of the mutants in the presence of TMAO.

The addition of TMAO to the PET assay in Figure 4b was intended to mimic the crowded cell cytosol in which Hsp90 functions in the in vivo experiments, as discussed in the first paragraph of pg 9 of the original submission. We chose TMAO to use as a crowder based on the work of our colleague Allen Minton (see ref #48 in the original submission) and because in our hands it is much easier to use than other crowdiers such as Ficoll. At Reviewer #1's suggestion, we prepared FOA plates containing 0.5M TMAO, based on Hassell et al. G3 2021. We repeated the shuffle experiment with the lethal mutants and were surprised to observe that the presence of TMAO must somehow interfere with FOA counterselection as the empty vector negative control grew robustly on FOA + TMAO, which precluded interpretation of any results. Thus, the effect of TMAO on rescuing the lethal mutants cannot be determined using this method.

Since TMAO and sorbitol behaved similarly in Hassell et al. (e.g., 0.5 M TMAO or 1 M sorbitol rescued high temperature growth defects caused by the dam1-1, dam1-9, duo1-2 or pkc1-1 alleles) we tested the idea in Reviewer #1's suggestion using sorbitol. Unlike TMAO, sorbitol did not have any deleterious effect on the FOA counterselection, since the empty vector did not grow but the wild type positive control grew well on FOA + sorbitol. We observed that sorbitol did not rescue any of the nonfunctional mutants, suggesting that failure to support

viability is not due to cell wall defects or other defects that may be overcome by addition of osmotic stabilizers to the medium. We present the results here but did not include them in the revised version:

Suggested revisions:

Discussion: The proposed structural role of the gamma phosphate of the nucleotide triphosphate reminds me of what has been proposed for the role of gamma phosphate of GTP in mediating subunit-subunit interactions within septin hetero-oligomeric complexes, where certain septins have independently evolved to become “pseudoGTPases” to restrict the kinds of subunit-subunit interactions that occur in vivo (see PMID 31990274, for example). Indeed, it seems that an Arg finger that is still present in some septins that are active GTPases was replaced during evolution by a His that contacts the gamma phosphate of a partner septin without triggering GTP hydrolysis (see PMID 38994454). If the authors also think this is a useful comparison, it may be worth mentioning.

The reviewer points to a study describing trans-interactions of arginine or histidine with GTP/GDP in different septin subunits that determines the organization of subunits in septin complexes. Interaction with histidine impairs hydrolysis while allowing, yet restricting, septin subunit interactions. We do find these findings interesting and that they would provide a useful comparison of hydrolysis vs conformational rearrangements that is worth mentioning. We added comparisons to it and other arginine fingers in the Discussion (pg 15, lines 9-18 and 21-25 of the revised version).

Abstract: Given the very broad readership of Nature Communications, it would be worth providing a tiny bit more context to the beginning of the paper, such as stating that Hsp90 is a highly conserved molecular chaperone that binds client proteins via a “clamp” to promote their native folding.

We added this sentence to the beginning of the Abstract in the revised version as suggested (pg 1, line 11): “Hsp90 is a highly conserved ATP-dependent molecular chaperone that forms a clamp around client proteins.”

The third sentence of the Abstract is also worded awkwardly: “mutating the conserved arginine R380, which interacts with the gamma phosphate of ATP in Hsp90’s closed clamp conformation, and other residues”. It is difficult to know if the “other residues” is the object of “mutating” or of “interacts with”. Please re-word.

We split the third sentence of the Abstract to clarify in the revision (pg 1, lines 13-16): “Here, we show that mutating a conserved arginine (R380 in budding yeast Hsp90) that interacts with the ATP γ phosphate in Hsp90’s closed clamp conformation changed Hsp90’s conformational response to ATP and ATP analogs. Mutating other residues in the nucleotide binding pocket have similar effects.”

Later in the Abstract: “ATP must properly align to interact with R380”. In the current wording, it is unclear with what ATP must properly align; is it something else, or is it R380?

We reworded the sentence in question (pg 1, 16-18): “These findings support the hypothesis that after binding, the ATP γ phosphate must reposition to interact with R380 and stabilize the closed clamp.”

The final sentence of the Abstract: “This in turns suggests that some arginine fingers might be structural elements important for regulation of inter-subunit or -domain interactions...”. The current wording could be construed as implying that this manuscript is making a novel suggestion that arginine fingers might regulate inter-subunit interactions apart from hydrolysis, when in fact this idea is already present in the literature (see above).

*There is a wealth of literature showing importance of arginine residues mediating inter-domain (or inter-subunit) interactions with nucleotide phosphates in ATP and GTP-dependent enzymes. Many assess roles for arginine in both hydrolysis and subunit interactions, so yes, these findings are already present. In many instances, however, uncertainties remain about whether the arginine plays a catalytic role or if the arginine is required to promote the conformational rearrangements needed for catalysis to occur. Unfortunately, in this statement we switched the focus from the essential structural role for ATP (our main conclusion stated in the title), to the arginine. We thank the reviewer for bringing attention to it. The point we wanted to make here is that the **nucleotide itself** is a structural component of some arginine fingers – apart from a source of energy – which to our knowledge has not been proposed before. The wording in the original submission does not convey this point, so in the revision we rewrote the final sentence of the Abstract (pg 1, lines 27-29): “This in turn suggests that for some arginine fingers the nucleotide itself is a structural element important for stabilization of inter-subunit or -domain interactions.”*

The first use of “yeast” on p. 2 line 15 should be specified as budding yeast or, better, *Saccharomyces cerevisiae*, since Hsp90 in fission yeast (for example) is not called Hsp82 and the Arg residue in question is R375, not R380.

We made the recommended change (pg 2, lines 16-17 of the revision).

p. 4 Line 17: “This idea is supported by the observation that the gamma-phosphate of ATP is not visible in the crystal structure of the Hsp90 N domain and is thus mobile”. I don’t quite follow the logic here. Is the idea that the crystal structure of the Hsp90 N domain is in neither the “open” nor the “closed” conformation, because the M domain is absent and therefore “open” or “closed” cannot be assigned, and the gamma-phosphate is mobile because it is not “locked” into one conformation or the other (“open” or “closed”)? I’m missing a logical step here about how the mobility of the gamma-phosphate in the N domain structure is relevant to the orientation that ATP must adopt in order for the clamp to close. Can the authors please spell this out a little better?

The idea is that the crystal structure of the N domain resembles the open state, since in the open state the N and M domains are separated. We have added to this sentence for clarification in the revision (pg 4, lines 21-24): “This idea is supported by the observation that the γ -phosphate of ATP is not visible, and is thus mobile, in the crystal structure of the Hsp90 N domain, which resembles the N domain in the open state of full length Hsp90 where the N and M domains are separated.”

p. 5 Line 16-17: “since cells transformed with plasmids encoding this mutation were unable to grow on medium containing 5'-fluoro-orotic acid (FOA)”: it might help the reader unfamiliar with this assay to add “URA3-marked” before “plasmids” here. Otherwise, it might seem that somehow FOA acts differently depending on whether Hsp40490 is wild-type or mutant per se.

We made clarifications in the revision (pg 5, lines 20-25): “As previously reported, Hsp82^{R380A} was unable to support viability, since cells lacking chromosomal Hsp90 genes that were transformed with a TRP1-marked plasmid encoding Hsp82^{R380A} could not lose the parental URA3-marked wild type Hsp90 plasmid during growth on uracil-containing medium and were subsequently killed by 5'-fluoro-orotic acid (FOA), which is toxic to cells expressing URA3 (Figure 1g; see Methods for details)”

Figure 1i: It would be very helpful here to include a mutant that *is* sensitive to CFW, to demonstrate that the CFW activity/concentration in the medium was sufficient for the intended purpose in this particular experiment.

We added in the revision a portion of a CFW plate (and YPD control) – cropped from the same plates used for the image in Figure 2c of the revision (2g of the original) – spotted with cells expressing E33A, which we showed previously is CFW^S and here serves as a control for CFW. The Figure legend was updated accordingly.

p. 6 line 6: “were mutated”: it seems that “are mutated” would make more sense here. Similarly, p. 6 line 8: “we chose residues that were conserved”: “are conserved” makes more sense, since they will always be conserved.

We made the suggested edits (pg 6, lines 15 & 18 of the revision).

p. 7 line 16: “or Sba1 or growth” should be “or Sba1 for growth”.

We made the correction (since this section was rewritten, the correction corresponds to pg 7, line 1 in the revision).

p. 8 lines 1-2: “the ability to form the closed clamp – with any nucleotide – is the prerequisite for supporting viability.” The “with any nucleotide” part here is a little confusing, since it would seem to refer to ATP γ S or AMPPNP, but the relevant assay is viability, and ATP γ S or AMPPNP will never be found in vivo.

(A similar concern was raised by Reviewer #2.)

The line that Reviewer #1 refers to was meant simply to highlight the correlation between the in vivo and in vitro results, namely that the mutants that supported viability all were able to close in the PET assay with at least one of the three nucleotides. The phrase “with any nucleotide” was included because the Hsp82^{E33A} protein, which is functional in vivo, closed only with ATP in PET assay, whereas all of the other functional (in vivo) mutant proteins closed with at least one of the analogs. It is also important to remember that wild type Hsp82 does not close

with ATP in the PET assay, but does with the non-hydrolysable analogs. Since the way these observations were originally presented was confusing, we rearranged the data in Figure 2 and rewrote the corresponding Results subsection to first present the in vivo data before introducing the PET data and to provide more clarity.

p. 13 line 28: “Some inviable Hsp82 mutant proteins”: it is a *very* minor point, but it is the mutant cells that are inviable, not the mutant proteins per se, so this should probably be reworded as “Some mutant Hsp82 proteins that did not support cell viability and did not form the closed clamp...” Also, in some places “non-viable” is used instead of “inviable”. I don’t have a strong preference, but the usage should be consistent throughout, to prevent confusion.

We reworded the sentence to provide clarity (pg 14, lines 25-27): “Some Hsp82 mutant proteins that did not form the closed clamp or support viability retained the ability to hydrolyze ATP, an observation reported for Hsp82^{R32A} and Hsp82^{R380A} previously.”

In the revision we define “functional” Hsp90 proteins as those that support viability (pg 5 lines 16-17) and refer them as functional, partially functional (e.g. can support viability but are slow and CFW^s) or nonfunctional throughout the revision to be more consistent.

Figure 5d,e and Supplemental Figure 5: it appears that the colonies/patches shown are sometimes from different plates, because there is an obvious “splicing” together of images. This makes sense, given that it would take a very large plate to accommodate so many strains side-by-side, and this is not a problem per se. However, I strongly recommend that the authors insert visible lines to indicate where images have been adjoined, and that in the figure legend, the authors indicate if the colonies/patches shown were grown on the same or on different plates. There may be some plate-to-plate variation (certainly we see this in our hands) and while it is not a major concern, it is best to be transparent about how the experiments were done.

We agree and made the suggested changes in the revision. If accepted, all uncropped original plate scans will be included in the supplementary data package, as well as a file detailing from which raw scan the images in each Figure were cropped.

Reviewer identity: Michael McMurray, PhD, University of Colorado School of Medicine

Reviewer #2 (Remarks to the Author):

The manuscript by Reidy and Masison reports on a large set of experiments to investigate the role of ATP in the function of Hsp90. Therefore, they use a combination of photon-induced electron transfer (PET), ATPase assays and various yeast viability studies. As far as I can judge, the experiments are well conducted.

The function of ATP and its hydrolysis in Hsp90 is an important question, which has already caught a lot of attention resulting in dozens of publications. In my view, many of the findings in this manuscript are either not new or largely speculation as detailed in the following.

The main conclusion of the manuscript is that the role of ATP in Hsp90 function is structural, namely that ATP is a severable linker that facilitates both the formation and dissociation of the closed clamp structure of Hsp90.

In my view, this is not much different from one of the conclusion of the following publication (which has obviously escaped the authors’ attention – at least it is not cited): Wolf et al., DOI: 10.1039/d0sc06134d . In this publication from 2021 it already has been shown that the hydrolysis of one ATP is coupled to a conformational change of Arg380, which in turn passes structural information via the large M-domain alpha-helix to the whole protein. It even describes

the stability of the closed clamp under various nucleotide conditions and the subsequent dissociation of the closed clamp.

As reviewer #2 points out, Wolf et al. 2021 reported numerous findings that are important to advancing the overall understanding of Hsp90 function. We have cited this work in the revision (pg 2 lines 19-20; pg 3, lines 19-21; pg 5 lines 2-3).

*That being said, Wolf et al. does not explicitly claim or even suggest that the primary role of the ATP molecule in Hsp90 function is to serve as a structural tether between the N and M domains. Rather, Wolf et al. examines the allosteric effects of ATP **hydrolysis** on Hsp90 structural dynamics using FRET experiments and MD simulations. The authors conclude that the hydrolysis of ATP has long-ranging allosteric effects on the structure of Hsp90. The point of our study was to examine the role of ATP **binding**, which unlike hydrolysis, is essential for in vivo function, i.e. cell viability. Our results led us to conclude that ATP acts as an essential structural component of Hsp90, not unlike certain small molecule cofactors that bind to enzyme domain interfaces and stabilize their interaction, such as the effect of inositol polyphosphates on HDAC activity.*

Altogether, I appreciate that with this new manuscript more evidence has been found for the structural role of ATP, but the main finding that ATP in Hsp90 has a structural role is not new. Therefore, altogether, I cannot support publication in Nature Communications.

Reviewer #2 states “the main finding that ATP in Hsp90 has a structural role is not new”, yet does not cite any study that explicitly claims that ATP binding is needed to close, as we do (pg 13 line 8 of the revision). Our idea that the structural stabilization of the closed clamp by ATP is the essential role of ATP in Hsp90 function is new and provides the long-sought explanation for the many findings and observations that Reviewer #2 refers to. Furthermore, we expand this idea by suggesting that the nucleotide component itself of arginine fingers, in certain cases, plays a structural rather than strictly energetic role as ATP does in Hsp90.

There are two studies that come close to suggesting what we conclude here, but the differences between them and our study are crucial. Cunningham et al. 2012 suggested that ATP stabilized the N and M domains through interaction with R380 but this interaction was to hydrolyze ATP more efficiently. Notably, the authors did not conclude that the essential role of ATP was to stabilize the closed clamp, as at that time the essential role of ATP hydrolysis as a source of energy to refold clients was unquestioned (until Zierer et al. 2016). A more recent publication from the Hugel group (Vollmar et al. 2024) suggested one of the essential roles of ATP hydrolysis was to reopen the Hsp90 dimer from the closed clamp. The conclusions from these studies support our model but are different than our work: they imply that ATP stabilizes the closed clamp but do not explicitly state or test the idea that ATP is a structural element as we have here. We appropriately cited both in the original submission (Cunningham et al.: pg 4 line 30; Vollmar et al.: pg 12 line 26), as well as clarified this point throughout the text in the revision.

Major points:

1) Result 1: “Positive charge at position 380 is necessary for closed clamp stabilization and in vivo Hsp90 function”. This has been shown before: See references: DOI: 10.1002/pro.2103 and DOI: 10.1039/d0sc06134d.

Cunningham et al. (ref #13 in the original submission) and Wolf et al. (see above) show that R380A does not form the closed clamp. In the original submission Cunningham et al. is appropriately cited (pg 12 lines 26-28 of the original submission) for this finding, as well as Zierer et al. (ref #27, pg 4 line 30). We also cite Wolf et al. for this finding in the revised version

(pg 5 lines 2-3). Indeed, we state that we used R380A as a non-closing control (pg 4 line 33 of original, pg 5 line 5 in revision) based on the earlier studies, which showed that loss of arginine was fatal for closed clamp formation. Those studies left open the question of whether positive charge from a different side chain (lysine) could complement this loss. The new finding in our study is that the R380K substitution forms the closed clamp and functions *in vivo* (like wild type). Thus, we show that it is the positive charge and not the arginine per se that is necessary for closed clamp and viability. We have added the words “and sufficient” to the subsection title in the revised version to clarify (pg 4, line 3).

2) Result 2: “Alterations in the nucleotide binding pocket result in different conformational responses to ATP and ATP analogs.” This is also clear from several previous studies, e.g. a very recent one in *Nat. Commun.* (which also seemed to have escaped the authors’ attention): <https://doi.org/10.1038/s41467-024-52995-y>

We are unaware of a study that looked at both mutations and different nucleotides in the manner we did, that is, to compare the combined effects of mutation AND nucleotide on Hsp90 conformation. Riedl et al 2024, the study Reviewer #2 mentions, investigates the differences the three nucleotides had on wild type yeast vs wild type human Hsp90 closing dynamics (e.g. in Figure 4), but does not combine the different nucleotides and mutational analyses. We have changed the subsection title to “Combined effects of nucleotide binding pocket mutation and nucleotide on Hsp90 conformation dynamics” (pg 6, lines 9-10).

In addition, I find it troublesome that there is no correlation between the effects the mutations had on the ability of the three nucleotides to stabilize closed clamp formation and phenotype.

*As Reviewer #1 points out (see above), the “relevant assay is viability”. The cell has the final say on whether any mutation impedes or kills Hsp90’s *in vivo* function. We interpret results from biochemical or other *in vitro* experiments through the lens of what supports viability. This is why we do not find it “troublesome” that the *in vivo* (i.e. physiological relevant) observations differ from what may be expected based on the results of biophysical experiments. Rather, the model must accommodate what the cells are telling us, even – or especially – if that means questioning long held underlying assumptions. In other words, the cells tell us what activities of Hsp90 are required for *in vivo* function and which are dispensable, and the *in vitro* experiments tell us what those activities are.*

How can the authors explain that the mutations that support viability sometimes show stabilization with ATP and in other cases with ATP analogs?

We proposed an explanation for these observations in the second paragraph of the Discussion: “altering the three dimensional topography of the nucleotide binding pocket changes the relative ability of the nucleotides to stabilize the closed clamp.”

*According to our model, *in vivo*, the γ -phosphate of the bound ATP must reorient to interact with R380. This reorientation is regulatable by co-chaperones, etc. as we show in Figure 4. This explains the observation that ATP does not induce Hsp82 to close in the absence of co-chaperones, crowder, etc. *in vitro*. The mutations alter the contours of the pocket and thus change the relative abilities of the nucleotides to interact with R380. This model explains why the mutants change the patterns in the PET assays relative to wild type. The mutations that support viability – even the ones which do not close with ATP *in vitro* (like wild type!) – must be able to close with ATP *in vivo*.*

In other words: Why is only stabilization with any one nucleotide necessary to see viability? In the cell, Hsp90 should only see ATP, therefore stabilization with ATP should be most important.

A similar concern was raised by Reviewer #1. It has been observed many times that in vitro ATP has little effect on stabilizing the closed clamp of wild type Hsp90, while the non-hydrolysable analogs readily stabilize the closed clamp. At first glance, this would seem to be a limitation of the PET assay (and related techniques), but instead this observation supports our idea that ATP-mediated closing is regulated in vivo. Indeed, we test this hypothesis and show it to be likely the case. Since ATP binding is essential, we proposed that after binding, ATP must reorient into a position that enables contact with R380. In the cell, this reorientation is regulated by co-chaperones and other factors, such as Sba1. The analogs are able to bypass this regulation. Altering the three dimensional topography of the nucleotide binding pocket (through mutations) changed the relative abilities of the nucleotides to stabilize the closed clamp in vitro. Again, despite the lack of closing with ATP in vitro (like wild type), mutant proteins that support viability must be able to close with ATP in vivo.

We added a sentence in the Results section to clarify this point: “Yet, ATP must induce closing in vivo since mutations that block closing or ATP binding are lethal.” (pg 4, lines 13-14)

3) Result 3: “E372K rescues growth defects of R32A and E33A and restores closed clamp stabilization by ATP- γ -S in vitro” – This in vitro result with an ATP analog is new.

It is unclear what Reviewer #2's concern is here.

4) Result 4: “Co-chaperones influence Hsp90's conformational dynamics differently in response to ATP or AMPPNP”. This has been shown before several times, e.g. also in <https://doi.org/10.1038/s41467-024-52995-y>.

*The concern here is similar to concern 2) above. The novelty of our work derives from the comparison of the effects of ATP vs. AMPPNP on Hsp90 closing in the presence of different co-chaperones (Sba1 or Sti1), which to our knowledge had not been reported. The experiments in Figure 4 investigated the **combined effects** of nucleotide (ATP vs. AMPPNP) **and** co-chaperone (Sti1 or Sba1). We cite Schmid & Hugel 2020 for the finding that Aha1 influenced Hsp90's dynamics with ATP in FRET assays, but Schmid & Hugel did not look at nucleotides other than ATP combined with Aha1. Likewise, Riedl et al. 2024 (now cited, pg 9, line 29; pg 9 line 34) investigates the effects of Aha1 or Sti1/Hop on yeast vs. human Hsp90 closing, but only with ATP- γ -S (Supplemental Figure 5 of Riedl et al.). We have reworded the subsection title: “Different co-chaperones influence Hsp90's conformational dynamics by either promoting or inhibiting ATP-R380 interaction” (pg 9, lines 20-21).*

In addition, here the authors write that AMPPNP is structurally predisposed to induced closed clamp formation, while before the authors' showed that stabilization by other nucleotides (ATP- γ -S, ATP) is also sufficient.

We assume Reviewer #2 is referring to pg 9 lines 16-18 (of the original submission): “These results lend support to our idea that the non-hydrolysable analog AMPPNP is structurally predisposed to induce closed clamp formation compared to ATP, as it was able to partially overcome the inhibitory effect of Sti1 on closed clamp formation.” This means that since Sti1 promoted a more open state upon addition of ATP, but addition of AMPPNP to Hsp82-Sti1 resulted in closing, we concluded that AMPPNP was able to overcome Sti1's inhibition of closing because it is “structurally predisposed” to interact R380 in spite of Sti1's best efforts to keep Hsp90 open, so to speak. We referred to AMPPNP specifically here because it was the analog

used in these experiments (since AMPPNP and ATP- γ -S had the same effect on wild type Hsp82, it could be considered redundant to include ATP- γ -S).

5) Result 5: "F120 regulates Hsp90 conformational dynamics by properly positioning R380". The data on the F120 mutation is very interesting, but I do not see any data on conformational dynamics.

The effect of F120A and F120Y on Hsp82's conformational dynamics via the PET assay are presented in Figures 2d (of the original submission, Figure 2j of the revision) and 5b, respectively.

6) Result 6: "Cdc37 coordinates Hsp90 conformational dynamics through F120". Again, I do not see any data on conformational dynamics.

A similar concern was raised by Reviewer #3.

The results of the experiments in Figure 6 show that Cdc37 and F120 interact genetically, and since changes at F120 impacted conformational dynamics (Figure 2d of original, 2j of revision; Figure 5b) and F120A was shown to impact kinase but not hormone receptor activity by the Bolon lab (i.e. F120A affected Cdc37 clients), we inferred that Cdc37 was regulating Hsp90's conformation dynamics. But, as Reviewer #2 correctly points out, there are no data directly reporting on conformational dynamics by Cdc37. We have changed the subsection title ("Cdc37 regulates multiple aspects of Hsp90's function in vivo" pg 10, line 17), a sentence in the Abstract ("Hsp90 residues E33 and F120 interacted genetically with the essential co-chaperone Cdc37, revealing Cdc37 regulates multiple aspects of Hsp90's in vivo functions." pg 1 line 22-23) as well as the last sentence of the first paragraph of the next subsection ("We reasoned that due to its proximity to R380 and Cdc37, F120 may play an important role in coordinating a functional interaction with Cdc37." pg 11 line 33 to pg 12 line 2) to clarify.

Minor comments:

- Title: I do not understand what the authors mean by "non-energetic"? I have not heard that word before. In my view, also the here described switch uses (transforms) energy, therefore it should be energetic.

We have removed the word "non-energetic" from the title in the revised version.

- P.13, line 2/3: "... Sti1 destabilizes the closed clamp even more when Hsp90 is bound to ATP than in the apo state...". Before, the authors wrote that there is not much closed clamp of Hsp90 (WT) in the ATP or apo state anyway. How does this fit together?

By "Before" we assume Reviewer #2 is referring to page 4, lines 1-2 of the original submission: "In the apo or ADP state the equilibrium is shifted mostly toward the open conformation." This is based on the work of Schulze et al. 2016, the study that first described the PET system we use here, which showed addition of ADP to purified Hsp90 (alone) in the PET assay did not significantly alter the fluorescence signal, that is: in the apo state Hsp90 is mostly open. We set "1" as the average fluorescence intensity of wild type in the apo state, which reflects an equilibrium of open and closed states, with the open state being more populated but not exclusive. The presence of Sti1 alone or with ATP shifts this equilibrium more towards the open side, resulting in a higher fluorescence (relative to wild type alone apo). Thus, it "fits together" by considering the effect of Sti1, which perhaps is more efficient at altering the apo equilibrium than ADP addition. We regret the confusion and have reworded some the

pertinent text in the Discussion to clarify: “Additionally, the reorientation of ATP is influenced by co-chaperones. Sti1 destabilizes the closed clamp in the absence of nucleotide and notably this effect is stronger when Hsp90 is bound to ATP. Sba1 has the opposite effect as Sti1 and stabilizes the closed clamp in response to ATP.” (pg 13, lines 21-24)

- Fig. 4: When are Sti1, Sba1 and TMAO added?

Co-chaperones and/or crowder were preincubated with Hsp82 prior to T=0. The Methods and Figure 4 legend have been updated to clarify (pg 18 line 26 and pg 28 line 9, respectively).

- For the PET measurements, the authors write “shown are the representative data sets of at least three independent experiments”. Does this mean that only one data set is shown? Or are all data sets are shown? Or is the average of the data sets shown? If only one data set is shown, what is the variation in the decay?

We have included all data points from 3 – 4 independent experiments to all the PET graphs in the revised version. We clarified the relevant text in the Methods section (pg 18 line 29-30).

Reviewer #3 (Remarks to the Author):

How exactly ATP binding and hydrolysis regulate the mechanism of Hsp90 has remained an open issue. This study suggests that ATP binding to Hsp90 plays an important structural role in the chaperone mechanism bringing different parts of the Hsp90 molecule together. The authors use an established PET-assay for Hsp82-NM docking together with experiments in yeast to study several mutants also in combination with co-chaperones and a molecular crowder. The find, that mutation of R380 and mutations close to the nucleotide binding pocket alter the formation of the closed clamp upon addition of ATP, ATP γ S and AMP-PMP. This supports the idea that ATP must be present in a specific orientation to drive conformational changes leading to the closed clamp conformation (tethering the N and M domains).

The orientation of ATP is also influenced by co-chaperones: Sti1 destabilizes the closed clamp whereas Sba1 stabilizes the closed clamp in response to ATP. It is suggested that Cdc37 regulates Hsp90 conformational dynamics by limiting or enhancing the influence of F120 on R380. The cellular environment/molecular crowding may shift the equilibrium further to the closed clamp formation.

Taken together, this is an interesting study that sheds new light on the consequences of ATP binding to Hsp90 and its regulation. Additional data reporting on nucleotide binding directly may further strengthen the study.

Specific points

1. ITC experiments may provide additional insight and direct evidence to what extent the mutants used in this study affect the affinity and thermodynamic parameters of nucleotide binding.

We designed our PET experiments to mitigate as much as possible the mutants' effects on affinity by having a 1000x excess of nucleotide (2 mM) over protein (2 μ M monomer). In this way we were able to study the changes the mutations had on conformational response to all three of the nucleotides we studied (ATP, AMPPNP and ATP- γ -S), each compared to wild type. Measuring the effect of the mutation on affinity would be a crucial part in understanding how any particular mutant resulted in closing with any particular nucleotide, but only together with much

more extensive biophysical experiments (like NMR) and molecular modeling. There are also technical considerations. We have tried ITC with Hsp82 in the past and due to the low native affinity for nucleotide ($K_D \sim 100 \mu\text{M}$, which requires very high concentrations of protein), the data do not neatly fit the models. Furthermore, our unpublished attempts at ITC agree with some findings in Minari et al. IJBM 2019, that the stoichiometry of ATP:Hsp90 dimer is not 2:1 as expected, suggesting another ATP binding site, and we do not at this time understand these findings and would be unsure how to interpret the results, especially considering the mutations combined with different nucleotides.

2. Figure 2c: The mutant G118A forms a closed clamp with ATP, also the mutant G123A with ATP and ATP_S but both are not viable. Is there an explanation for this?

A similar concern was raised by other reviewers, see above. We have added new data in the revision that shows G118A and G123A have dominant negative growth defects. Cells expressing both wild type Hsp82 and either G118A or G123A grow significantly slower than cells expressing both wild type Hsp82 and an empty vector or any of the lethal non-closing mutants (Figure 2b in the revision). We have also rearranged the data in Figure 2 of the revised version (and corresponding Results section) to present the in vivo functional assays before the PET data, in order to facilitate understanding of the various correlations among all the data points.

Thus, the mutants fall into one of three groups:

- 1. Functional mutations, i.e., those that support viability. All of these are able to close with at least one of the nucleotides in the PET assay.*
- 2. Nonfunctional mutations that do not close with any of the three nucleotides tested in the PET assay. These mutants do not exhibit a dominant negative growth phenotype.*
- 3. Nonfunctional mutations that do close with at least one of the nucleotides in the PET assay. These mutants have a dominant negative growth phenotype.*

The mutants in the third group, G118A and G123A, are most likely defective in Hsp90 activities downstream of the ability to close. These defects are so severe that they interfere with the function of wild type Hsp90, either through mixed-dimer formation (mutant and wild type), on their own (mutant homodimers) or both. Our observations generally agree with a recent study (Flynn, et al. 2025) that specifically investigated dominant negative effects of Hsp90 mutants including G123H. Defining the mechanism by which G118A and G123A cause the dominant negative phenotype is somewhat beyond the scope of the present study and was adequately addressed by Flynn et al. We therefore added citation of Flynn et al. in the Discussion of the of the revised version (pg 13 line 32 to pg 14 line 7). Note, citing Flynn et al. was also a specific request of this reviewer, see below.

3. Would the PET-docking experiment with Cdc37 and Cdc37_S14A together with Hsp82, E33A and F120A lead to the same conclusions as proposed based on the in vivo assays in figure 6?

We have attempted PET assays with Cdc37 but see no effect of including this protein (or S14A or S14E, etc. mutants) in the reaction. We think those results are explained by findings from the Agard lab that showed that Cdc37 itself undergoes conformational shifts upon binding to the kinase client, prior to interaction with Hsp90. Cdc37 apparently does not interact with Hsp90 unless it is first bound to client. Unfortunately, despite vast (and ongoing) efforts we have not been able to produce usable amounts of a soluble kinase client for use in the PET assay. This type of experiment that combines Hsp90, co-chaperone and client has been and remains a goal for us. It is for these reasons that we investigated the combinations of Cdc37 and Hsp90 mutants in vivo, and we created strain MR1154 to do so. As Reviewer #1 states above, “the

relevant assay is viability”; however, we agree that the genetic data do not directly report on the conformational dynamics (a similar concern was raised by Reviewer #2). Therefore, we have retitled the subsection (“Cdc37 regulates multiple aspects of Hsp90’s function in vivo”) and reworded a sentence in the Abstract (“Hsp90 residues E33 and F120 interacted genetically with the essential co-chaperone Cdc37, revealing Cdc37 regulates multiple aspects of Hsp90’s in vivo functions.”) in the revision to clarify.

4. A recently published paper on loss-of function and dominant negative mutations has some overlap with the mutants used in this work: Flynn, Julia M., Margot E. Joyce, and Daniel N. A. Bolon. 2025. This study should be included in the discussion

See item 2 above. The new data in the revision regarding the dominant negative phenotypes of the inviable mutants that retained the ability to close in the PET assay (G118A and G123A) afforded the perfect opportunity to discuss Flynn et al., and we do so in the revision Discussion (pg 14 lines 2-7).

5. Statistical significance should be determined for numerical data shown in the various bar graphs.

We have included statistical analyses in all of the bar graphs in the revision and updated the Methods and Figure legends where appropriate.

Minor

For direct comparison, the ATP γ S traces of figures 3c and 2a and 3d and 1b should be combined in one figure.

We added the R32A + ATP γ S fit line from Figure 2a to Figure 3c and the E33A + ATP γ S fit line from Figure 1b to Figure 3d (blue dashed lines) to aid in direct comparison. The Figure legend was also updated.

In figure 1f, “rates” is lower case, whereas in figure 1h it is upper case.

We made the correction.

Figures 3 and 4: The font size is much larger than in the other figures.

This is due to pasting in screenshots of the figures into the manuscript for the initial submission (Figures 3 and 4 are single column figures where the rest are 2 column). We have ensured consistent font sizes in the final figures.

Figure 5C: for the mutant F120Y only duplicates are shown.

We corrected this oversight.

Page 2, line 11: “co-called” - “so-called”

We made the correction.

Response to Reviewers 2

We thank the reviewers for their time and attention in reviewing the revised version of our manuscript. Changes in the second revision are mostly edits done at the guidance of the editorial staff, for example a rewrite of the Abstract to shorten it to adhere to the journal's policies, slight changes to the last paragraph of the Introduction, removal of excessive italics, etc. The changes made at the Reviewers' request are outlined in detail below. Changes we made that were not at the reviewers' or editor's suggestion was to remove images of cells expressing a mutation that was not discussed anywhere else in the manuscript (A107N) from Figure 6b and Supplemental Figure 2b&c, whose inadvertent inclusion had been overlooked (by us and the reviewers). The Reviewers' comments are pasted below, and our point-by-point response is in *italics*. All page and line numbers refer to the second revision.

Reviewer #1 (Remarks to the Author):

I am satisfied with the authors' revisions to the manuscript and how they address my and other reviewer concerns.

Reviewer #2 (Remarks to the Author):

The authors have substantially improved their manuscript. They have incorporated new data on mutations, enhanced the integration of their findings within the existing literature, and refined the comparison between *in vivo* and *in vitro* results.

As outlined in my initial review and elaborated upon below, I still do not fully agree with all aspects of the data interpretation or the claimed novelty. However, given the central importance of ATP in Hsp90 function, it is essential to allow for multiple perspectives. Therefore, I now support the publication of this manuscript, though I believe the following points should still be addressed:

1. Many researchers, including myself, have been influenced by the extensive body of work on molecular motors such as myosins and kinesins, where ATP binding, hydrolysis, and phosphate release are well-established as structural drivers of the mechanochemical cycle. In that context, it seemed intuitive that each step has a structural function in ATPases. Nevertheless, I agree that explicitly stating this for Hsp90 is important. The revised wording, particularly the removal of the term "non-energetic," represents a clear improvement, but maybe the authors can even further clarify their wording with this information.

We believe our description of the structural effects on ATP binding and hydrolysis on Hsp90 on pg 12, lines 28-34 (ending with: "Thus, Hsp90 needs ATP to close.") and pg 14 lines 1-5 is sufficiently explicit. We do discuss the broad implication of our model in the context of AAA+ ATPases and other arginine finger systems, therefore we do not think it is necessary to include more text to an already lengthy Discussion to mention myosins and kinesins.

2. I acknowledge that the authors' approach to examine these mutations, cochaperones, and different nucleotides in this combination is novel. However, several of these individual components and their interactions have been studied previously. I cannot provide a comprehensive literature review here and I appreciate that this also cannot be done for this manuscript, but for example, the following publication investigated the effects of Sti1 and Sba1 in the context of ATP and AMP-PNP (see DOI: 10.1038/nsmb.1965), i.e. some of the main components in this work.

We do not claim to be the first to study Sti1 or Sba1, we simply state that we included them in our PET assays (pg 9 lines 20-21). The study Reviewer #2 refers to (now cited) does indeed investigate the roles of Sti1 and Sba1 (aka p23) in Hsp90 complex formation and in some experiments uses AMPPNP and in others uses ATP. For example, in Figure 1a of that study the authors report the inhibition of Hsp90 ATPase by Sti1, while in the experiments in Figure 3 the complexes formed between Hsp90, Sti1, p23 and other cochaperones were analyzed in the presence of AMPPNP. But the authors of that study did not explicitly investigate the different effects caused by AMPPNP compared to ATP, as we did. We realize that our point may be a fine hair to split, but nevertheless we remain confident that our approach - investigating the different effects of the nucleotides - is distinct, as Reviewer #2 acknowledges.

3. I find the following new sentence problematic: "ATP must induce closing in vivo since mutations that block closing or ATP binding are lethal." While this might be true, it should be expressed more cautiously, because strict causality is not given here. For instance, the mutations might inhibit closing through mechanisms independent of ATP, or the inability to bind ATP could impair other critical steps (such as lid closure) rather than directly preventing N-terminal domain closure.

We changed the sentence slightly: "Yet, in all likelihood ATP induces closing in vivo since mutations that block either closing or ATP binding are lethal." (pg 4 lines 2-4).

Reviewer #3 (Remarks to the Author):

The authors have answered the queries raised by me in detail. I appreciate their comments on determining the affinity of ATP and Hsp90 by ITC - and the description of the results of their efforts in using ITC for Hsp90 analysis. It is unfortunate that there seem to be technical issues that preclude the use of ITC in this context. The authors may nevertheless comment on the effects of mutations on the affinity of ATP binding and how this may or may not affect the results obtained.

We added this sentence in the Results (pg 8 lines 3-6): "While our experimental designs intentionally minimized possible effects the mutations had on nucleotide affinity, we cannot rule out that some of our observations are influenced by alterations in affinity."

The other points raised were answered in a satisfactory manner. A general comment: I think split ATPases (GHKL ATPases) are in general constructed in a way that ATP binding induces conformational changes and that hydrolysis then resets the system.